# Greedy Importance First (GIF): Importance-Aware Scheduling for Hyperparameter Optimization

## Abstract

Hyperparameter Optimization (HPO) is essential for building high-performing ML/DL models, yet conventional optimizers often struggle in high-dimensional spaces where evaluations are costly and progress is diluted across many low-impact variables. We propose Greedy Importance First (GIF), an importance-aware scheduling strategy that uses a small-sample warm start to estimate per-hyperparameter importance, forms importance-driven groups, allocates budget proportionally, and retains a full-space fallback. Under fixed evaluation budgets, we study GIF on diverse benchmarks—five anisotropic high-dimensional analytic functions ($d \in \{5, 10, 30, 50\}$), Bayesmark, and NAS-Bench-301 (33D). GIF consistently attains faster convergence and stronger final incumbents than baselines (TPE, BOHB, Random Search, and Sequential Grouping) in higher-dimensional settings; on Bayesmark, where the effective dimensionality is smaller, GIF remains competitive, but the margins are modest. Ablations confirm the value of importance estimates, proportional allocation, and the full-space fallback. Our Hyperparameter Importance Assessment (HIA) also recovers the intended anisotropy on those anisotropic analytic functions. Overall, GIF offers a simple, plug-compatible approach for more sample-efficient HPO in high-dimensional spaces, with potential relevance to deep-model tuning and large-scale AutoML.

## 1 Introduction

Hyperparameter optimization (HPO) is a critical stage in modern ML/DL pipelines: it governs robustness, stability, and generalization. Despite a mature toolbox—Bayesian optimization (e.g., TPE (Bergstra et al., 2011), BOHB (Falkner et al., 2018)), evolutionary (Loshchilov & Hutter, 2016), and bandit methods (Li et al., 2018)—efficiency often degrades as dimensionality grows: each evaluation becomes costlier and surrogates become harder to fit and less informative (Bischl et al., 2023). Crucially, the obstacle is not dimensionality alone but the strongly uneven influence of hyperparameters (Probst et al., 2019). In many models, a small subset of settings accounts for most performance variation, while others contribute marginally. Yet most optimizers advance all coordinates in lockstep each iteration, effectively enforcing uniform scheduling. This induces a dimensionality bottleneck: treating all hyperparameters equally dilutes the budget and delays progress, especially under tight evaluation limits.

Hyperparameter importance assessment (HIA) provides a principled foundation for addressing this bottleneck: from a small set of trials, it estimates each hyperparameter's marginal contribution to performance—and, when needed, pairwise interactions. However, despite the availability of estimators such as N-RReliefF (Wang et al., 2024), fANOVA (Hutter et al., 2014), and PED-ANOVA (Watanabe et al., 2023), there is no widely adopted strategy that operationalizes these estimates into concrete scheduling decisions. As a result, HIA methods are underutilized in practice.

This paper introduces Greedy Importance-First (GIF), an importance-aware HPO strategy that turns HIA insights into an explicit, budgeted search plan. As illustrated in Fig. 1, GIF (i) performs a small-sample warm start to collect initial trials and produce insights interactively with HIA algorithms; (ii) orders hyperparameters by estimated importance and groups them accord-

ingly; (iii) allocates budgets proportionally to group importance and optimizes each group while fixing other variables at the current incumbent, warm-starting from the accumulated history; and

(iv) when a round yields no improvement, falls back to joint optimization to restore global exploration. This design concentrates the budget where it matters most, while the fallback to joint optimization provides a principled escape from local stagnation. We evaluate GIF under fixed budgets on controlled anisotropic analytic functions, Bayesmark

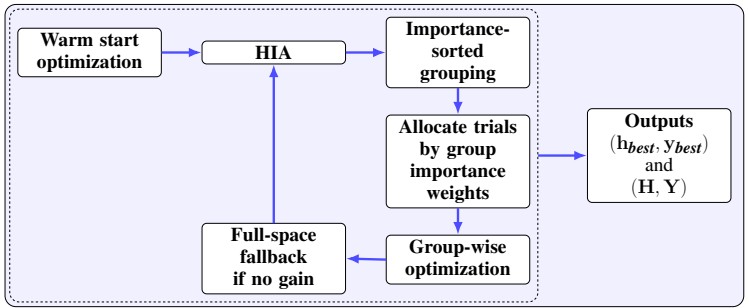

Figure 1: **GIF Pipeline:** High-level workflow of the proposed Greedy Importance First strategy.

tasks (Turner & Eriksson, 2019), and NAS-Bench-301 (Zela et al., 2020). Ablations disentangle the effect of each component, and we further verify that HIA can recover the ground-truth anisotropy on the analytic benchmarks—even with limited evaluations, it correctly highlights the few dominant coordinates while suppressing negligible ones.

**Contributions:**

- We propose *Greedy Importance First* (GIF), which turns HIA into a concrete search plan: importance-sorted grouping, importance-proportional allocation, and a safeguarded full-space fallback.
- We verify that lightweight HIA (N-RReliefF) recovers the intended anisotropy on controlled functions, supporting its use as a reliable prior under tight budgets.
- Under fixed budgets, GIF is consistently competitive and often superior in higher dimensions (weighted analytic functions, NAS-Bench-301), while remaining competitive on mid-dimensional Bayesmark; ablations confirm each component's contribution.

## 2  RELATED WORK

**High-dimensional BO.** High dimensionality stresses both surrogate modeling and acquisition optimization. Prior work mitigates this via (i) subspace or variable-selection assumptions (Wang et al., 2016; Letham et al., 2020; Nayebi et al., 2019), (ii) additive kernel decompositions (Ngo et al., 2025), and (iii) local/trust-region BO (Eriksson et al., 2019b), all of which effectively reduce model complexity. A recent reassessment argues that a principal failure mode of "vanilla BO" in high-dimensional settings is excess assumed complexity, and shows that simple length-scale prior scaling can render standard BO strongly competitive without imposing structural low-dimensional assumptions (Hvarfner et al., 2024). Our approach is orthogonal: rather than constraining the function class or geometry, GIF reallocates evaluation budget using empirically inferred (HIA-based) hyperparameter importance, and retains a full-space fallback to preserve global exploration.

**Hyperparameter Importance Assessment (HIA).** Understanding which hyperparameters "matter" has long supported post-hoc analysis and space design; for example, Weights & Biases (W&B) Sweeps provide importance plots from trial histories (Weights & Biases, 2025), while libraries such as Optuna and SMAC3 expose fANOVA-based importance tools (Akiba et al., 2019; Lindauer et al., 2022). Methodologically, fANOVA remains a standard variance-decomposition approach (Hutter et al., 2014); PED-ANOVA generalizes it with a Pearson-divergence–based closed form that enables efficient local importance on arbitrary subspaces (e.g., top-performing regions) (Watanabe et al., 2023). Complementary to fANOVA-style decompositions, N-RReliefF adapts ReliefF to continuous responses and quantifies both marginal and pairwise interaction importance from HPO histories, offering a lightweight, data-driven estimator under tight budgets (Wang et al., 2024).

**Gray-box and uncertainty-aware HPO.** Gray-box approaches enrich BO surrogates with intermediate training signals (e.g., learning curves, checkpoint features, or partial-fidelity measurements), and uncertainty-aware schedulers couple candidate selection with budget allocation to avoid premature discarding under early-stage noise (Liu et al., 2024; Mehta et al., 2024; Falkner et al., 2018).

While these methods exploit richer signals across candidates, GIF serves as a lightweight allocator across hyperparameters: it uses HIA from small warm-starts to reweight search effort across hyperparameters and can plug into BOHB/TPE-style optimizers as the inner engine.

**Resource allocation, warm starts, and scheduling.** Many HPO systems exploit warm starts (e.g., transferring priors or surrogate states), parallel scheduling, or multi-fidelity allocation across candidates and tasks (Wistuba et al., 2018; Falkner et al., 2018; Li et al., 2018; Swersky et al., 2013; Wang et al., 2025). However, they typically retain uniform treatment across hyperparameters within an iteration. GIF breaks this per-iteration uniformity by fixing non-targeted hyperparameters to the current incumbent and concentrating trials on the most important groups. This increases the signal-to-noise ratio per evaluation in high-dimensional regimes. A per-round full-space fallback then provides a principled escape hatch from local plateaus.

In sum, GIF turns early HIA into a concrete search plan: importance-ordered grouping, importance-proportional allocation, warm-started subspace search, and a safeguarded full-space fallback. This yields a plug-compatible route to sample-efficient HPO in high-dimensional settings, complementary to structural high-dimensional BO (subspace/variable-selection assumptions and local/trust-region BO), gray-box surrogates, and uncertainty-aware schedulers.

## 3 PROBLEM SETUP

We consider HPO on a fixed dataset $\mathcal{D}$ and hyperparameter search space $\Theta = \Theta_1 \times \cdots \times \Theta_d$, where each $\Theta_i$ is the domain of hyperparameter $H_i$. A configuration is $\mathbf{h} = (h_1, \ldots, h_d) \in \Theta$. The black-box objective is $f_{\mathcal{D}} : \Theta \to \mathbb{R}, \mathbf{h} \mapsto f_{\mathcal{D}}(\mathbf{h})$, which returns a scalar performance (e.g., validation accuracy to maximize). The goal of HPO is $\mathbf{h}^{\star} \in \arg\max_{\mathbf{h} \in \Theta} f_{\mathcal{D}}(\mathbf{h}), y^{\star} = f_{\mathcal{D}}(\mathbf{h}^{\star})$, subject to a limited evaluation (or wall-clock) budget $B_{\text{total}}$. After $t$ evaluations, the history is $\mathcal{H} = \{\mathbf{h}^{(1)}, \ldots, \mathbf{h}^{(t)}\}$ and $\mathcal{Y} = \{y^{(1)}, \ldots, y^{(t)}\}$ with $y^{(i)} = f_{\mathcal{D}}(\mathbf{h}^{(i)})$. The incumbent (best-so-far) configuration is $(h_{\text{best}}, y_{\text{best}})$ where $y_{\text{best}} = \max_{i \leq t} y^{(i)}$. An optimizer $\mathcal{A}_{\text{opt}}$ proposes new candidates conditioned on $(\mathcal{H}, \mathcal{Y})$, evaluates them, and appends results until $B_{\text{total}}$ is exhausted. Standard outputs are the final incumbent $(h_{\text{best}}, y_{\text{best}})$ and the complete trace $(\mathcal{H}, \mathcal{Y})$.

**Representative baseline.** TPE (Bergstra et al., 2011) partitions the history $(\mathcal{H}, \mathcal{Y})$ by a score threshold $y_0$ (e.g., the $\gamma$-quantile), and fits conditional densities $l(\mathbf{h}) = p(\mathbf{h} \mid y \geq y_0)$ and $g(\mathbf{h}) = p(\mathbf{h} \mid y < y_0)$. New candidates maximize $l(\mathbf{h})/g(\mathbf{h})$, a proxy for expected improvement. In practice, $l$ and $g$ are estimated via Parzen windows with the factorization $p(\mathbf{h}) \approx \prod_{j=1}^{d} p(h_j)$. The iterative loop is: fit densities $\to$ sample $\mathbf{h}^{(t+1)} \to$ evaluate $f_{\mathcal{D}}(\mathbf{h}^{(t+1)}) \to$ update the history.

**Typical bottlenecks in High Dimensions.** For the representative optimizer TPE, as the dimensionality $d$ of the search space increases, several limitations arise under tight evaluation budgets $B_{\text{total}}$: **(i)** The independence assumption $p(\mathbf{h}) \approx \prod_j p(h_j)$ neglects coordinate interactions. In high-dimensional hyperparameter spaces, many variables only matter through their joint effects. Ignoring such dependencies causes both $l(\mathbf{h})$ and $g(\mathbf{h})$ to appear nearly uniform across most coordinates, offering little guidance for exploration. **(ii)** In higher dimensions, density estimation becomes increasingly noisy because the effective sample size per coordinate shrinks. With limited evaluations, each marginal distribution is poorly supported, so $l(\mathbf{h})$ and $g(\mathbf{h})$ fluctuate heavily, yielding unstable search guidance. **(iii)** As $d$ increases, contributions of coordinates to $f$ are highly imbalanced; the presence of many low-impact dimensions reduces the effective signal-to-noise in each sampled evaluation, leading to slower improvement over iterations. These effects explain why traditional BO (here represented by TPE) struggles in high dimensions, motivating importance-aware reallocation strategies. To address the high-dimensional bottlenecks of standard BO, we introduce hyperparameter importance assessment (HIA) as guiding insights, which GIF leverages to order, group, and allocate evaluation budgets more effectively.

## 4 THE GIF ALGORITHM

### 4.1 PIPELINE OVERVIEW

Algorithm 1 formalizes how the key components of GIF are orchestrated into a single scheduling strategy.

---

**Algorithm 1** GIF Main Strategy

---

**Require:** Search space $\Theta$, objective $f_{\mathcal{D}}$, subsample ratio $\alpha$, initial budget $B_{\text{init}}$, step size $B_{\text{step}}$, total budget $B_{\text{total}}$, max group size $k$, importance evaluator $\mathcal{A}_{\text{imp}}$, optimizer $\mathcal{A}_{\text{opt}}$, fallback ratio $\rho$

**Ensure:** Incumbent $(\mathbf{h}_{\text{best}}, y_{\text{best}})$, the complete evaluation trace $(\mathcal{H}, \mathcal{Y})$

1: $(\mathcal{H}, \mathcal{Y}), (\mathbf{h}_{\text{best}}, y_{\text{best}}) \leftarrow$ **warm-start**$(\Theta, f_{\mathcal{D}}, \alpha, B_{\text{init}}, \mathcal{A}_{\text{opt}})$ (see Alg. 3)
2: $T_{\text{used}} \leftarrow B_{\text{init}}, \quad T_{\text{full used}} \leftarrow 0, \quad B_{\text{full total}} \leftarrow \rho B_{\text{total}}$
3: **while** $T_{\text{used}} < B_{\text{total}}$ **do**
4: $\quad I \leftarrow \mathcal{A}_{\text{imp}}(\mathcal{H}, \mathcal{Y})$ {importance weights $\{I_i\}_{i=1}^d$}
5: $\quad$**FormGroups:** sort indices by $I$ (desc.), then partition into groups $\mathcal{G} = \{\mathcal{G}_j\}$ with $|\mathcal{G}_j| \leq k$
6: $\quad B_{\text{cur}} \leftarrow \min(B_{\text{step}}, B_{\text{total}} - T_{\text{used}})$
7: $\quad \mathbf{b} \leftarrow$ **AllocateBudget**$(\mathcal{G}, I, B_{\text{cur}})$ (see Alg. 4)
8: $\quad (\mathcal{H}, \mathcal{Y}, h_{\text{best}}, y_{\text{best}}, T_{\text{used}}, improved) \leftarrow$
$\quad\quad$ **GroupOpt**$(\mathcal{G}, \mathbf{b}, \mathcal{H}, \mathcal{Y}, h_{\text{best}}, y_{\text{best}}, T_{\text{used}}, \mathcal{A}_{\text{opt}})$ (see Alg. 5)
9: $\quad$**if** not $improved$ **and** $T_{\text{full used}} < B_{\text{full total}}$ **and** $T_{\text{used}} < B_{\text{total}}$ **then**
10: $\quad\quad R \leftarrow \left\lfloor \frac{B_{\text{total}} - T_{\text{used}}}{B_{\text{step}}} \right\rfloor + 1$
11: $\quad\quad B_{\text{full}} \leftarrow \min\left(\left\lfloor \frac{B_{\text{full total}} - T_{\text{full used}}}{R} \right\rfloor, B_{\text{total}} - T_{\text{used}}\right)$
12: $\quad\quad (\mathcal{H}, \mathcal{Y}, h_{\text{best}}, y_{\text{best}}, T_{\text{used}}, T_{\text{full used}}) \leftarrow$ **FullSpaceOpt**$(\Theta, f_{\mathcal{D}}, B_{\text{full}}, \mathcal{H}, \mathcal{Y}, h_{\text{best}}, y_{\text{best}},$
$\quad\quad\quad\quad\quad\quad\quad\quad\quad\quad\quad\quad\quad\quad\quad\quad\quad\quad\quad T_{\text{used}}, T_{\text{full used}}, \mathcal{A}_{\text{opt}})$ (see Alg. 6)
13: $\quad$**end if**
14: **end while**
15: **return** $(\mathbf{h}_{\text{best}}, y_{\text{best}})$ and $(\mathcal{H}, \mathcal{Y})$

---

## 4.2 WARM START

**Inputs:** Search space $\Theta$, dataset $\mathcal{D}$ (size $|\mathcal{D}|$), objective $f_{\mathcal{D}} : \Theta \to \mathbb{R}$, subsample ratio $\alpha \in (0, 1]$, warm-start budget $B_{\text{init}}$, inner optimizer $\mathcal{A}_{\text{opt}}$. **Outputs:** Initial history $(\mathcal{H}, \mathcal{Y})$ with $|\mathcal{H}| = |\mathcal{Y}| = B_{\text{init}}$, and incumbent $(\mathbf{h}_{\text{best}}, y_{\text{best}})$. We randomly subsample the dataset to obtain $\mathcal{D}_{\text{init}}$ of size $\alpha|\mathcal{D}|$ and run $\mathcal{A}_{\text{opt}}$ for $B_{\text{init}}$ evaluations on $\Theta$ (using $\mathcal{D}_{\text{init}}$), producing $(\mathcal{H}, \mathcal{Y})$ and initializing $(\mathbf{h}_{\text{best}}, y_{\text{best}})$ as the best in this history. This warm start expends a small budget to gather optimizer-guided (rather than purely random) configurations—reducing wall-clock cost via subsampling while providing a more diverse, informative basis for subsequent importance estimation under small budgets. Further pseudocode is given in App. B, Alg. 3.

## 4.3 HYPERPARAMETER IMPORTANCE ASSESSMENT (HIA)

**Inputs:** Optimization history $(\mathcal{H}, \mathcal{Y})$; evaluator $\mathcal{A}_{\text{imp}}$. **Outputs:** Normalized importance profile $\{I_i\}_{i=1}^d$ assigning a nonnegative weight to each hyperparameter $H_i$.

In general, HIA methods assign weights $I_1, \ldots, I_d$ estimating each hyperparameter's marginal contribution to performance, providing interpretable insights about "what matters" and informing downstream scheduling or search-space design. Representative techniques include *fANOVA* (Hutter et al., 2014), *PED-ANOVA* (Watanabe et al., 2023), and *N-RReliefF* (Wang et al., 2024). In GIF, we employ *N-RReliefF* as our default importance evaluator. Given history $(\mathcal{H}, \mathcal{Y})$, N-RReliefF treats each configuration as a reference point, compares it with its nearest neighbors in configuration space, and accumulates per-dimension covariation weighted by the performance difference between neighbors. This produces raw scores $\widehat{I}_i$, which are then mapped into positive, comparable importances via a softplus normalization and re-scaled so that $\sum_i I_i = 1$ (see details in App. A). In this way, dimensions where small input changes consistently lead to large performance shifts are assigned higher weights, which in turn underpin ordering and grouping in GIF.

## 4.4 GROUPING AND ALLOCATION

**Key Inputs:** Importance weights $\{I_i\}_{i=1}^d$; maximum group size $k$; per-round step size $B_{\text{step}}$; total budget $B_{\text{total}}$; used trials $T_{\text{used}}$. **Outputs:** A partition of hyperparameter indices into groups $\mathcal{G} = \{\mathcal{G}_j\}$ with $|\mathcal{G}_j| \leq k$, and per-group trials allocations $\mathbf{b} = [b_1, \ldots, b_{|\mathcal{G}|}]$. We first set the current

round budget $B_{\text{cur}} = \min\big(B_{\text{step}}, \ B_{\text{total}} - T_{\text{used}}\big)$. Based on $\{I_i\}$, we sort hyperparameters by descending weight and partition them into groups of size at most $k$. For each group $\mathcal{G}_j$, we compute its total weight $I_j = \sum_{i \in \mathcal{G}_j} I_i$ and allocate trials proportionally: $b_j = \max\Big(1, \ \big\lfloor \frac{I_j}{\sum_g I_g} B_{\text{cur}} \big\rfloor\Big)$. Enforcing $b_j \geq 1$ guarantees at least one trial per group; the final allocations $\mathbf{b}$ are then passed to the group-wise optimization stage.

## 4.5 GROUP-WISE OPTIMIZATION

**Key Inputs:** Groups $\mathcal{G} = \{\mathcal{G}_j\}$, per-group allocations $\mathbf{b} = [b_1, \ldots, b_{|\mathcal{G}|}]$, current history $(\mathcal{H}, \mathcal{Y})$, incumbent $(\mathbf{h}_{\text{best}}, y_{\text{best}})$, and inner optimizer $\mathcal{A}_{\text{opt}}$. **Outputs:** Updated history $(\mathcal{H}, \mathcal{Y})$, updated incumbent $(\mathbf{h}_{\text{best}}, y_{\text{best}})$, and updated trial counter $T_{\text{used}}$.

For each group $\mathcal{G}_j$, we fix all hyperparameters outside $\mathcal{G}_j$ to their values in the current incumbent $\mathbf{h}_{\text{best}}$. We then invoke the inner optimizer $\mathcal{A}_{\text{opt}}$ for $b_j$ evaluations restricted to $\mathcal{G}_j$, with `warm-start` from the existing history $(\mathcal{H}, \mathcal{Y})$. The resulting evaluations $(\mathcal{H}_j, \mathcal{Y}_j)$ are appended to the history, and $T_{\text{used}}$ is incremented by $b_j$. After each group is optimized, we update the incumbent if a better configuration is discovered. If all groups fail to improve the incumbent, the round is considered *unsuccessful*, potentially triggering the full-space fallback (4.6). Otherwise, the algorithm proceeds with the next round using the updated history and incumbent.

## 4.6 FULL-SPACE FALLBACK

**Key Inputs:** Remaining trials $T_{\text{left}} = B_{\text{total}} - T_{\text{used}}$; full-space reserved quota $B_{\text{full total}} = \rho B_{\text{total}}$; cumulative full-space trials used $T_{\text{full used}}$ (i.e., trials already spent on full-space fallback); step size $B_{\text{step}}$. **Outputs:** updated evaluation history $(\mathcal{H}, \mathcal{Y})$ and updated incumbent $(\mathbf{h}_{\text{best}}, y_{\text{best}})$. To guard against subspace stagnation while balancing exploration–exploitation, GIF triggers a full-space step *only* when an entire group-wise round yields no improvement. Given $T_{\text{left}}$, define the remaining full-space quota $T_{\text{full left}} = \max\big(0, \ B_{\text{full total}} - T_{\text{full used}}\big)$, and the estimated number of future rounds $n_{\text{round}} = \big\lfloor \frac{T_{\text{left}}}{B_{\text{step}}} \big\rfloor + 1$. Allocate a per-round fallback budget $B_{\text{full}} = \min\big( \lfloor T_{\text{full left}}/n_{\text{round}} \rfloor, \ T_{\text{left}} \big)$. Run the inner optimizer on the full space $\Theta$ for $B_{\text{full}}$ evaluations with warm-start $(\mathcal{H}, \mathcal{Y})$, obtain $(\mathcal{H}_{\text{full}}, \mathcal{Y}_{\text{full}})$, and update $(\mathbf{h}_{\text{best}}, y_{\text{best}})$, $T_{\text{used}} \leftarrow T_{\text{used}} + B_{\text{full}}$, $T_{\text{full used}} \leftarrow T_{\text{full used}} + B_{\text{full}}$. If group-wise optimization keeps improving, the fallback is never activated; the algorithm continues with the standard per-round budget until $T_{\text{used}} = B_{\text{total}}$, and any unused full-space quota remains unspent.

**Implementation Note** The inner routine $\mathcal{A}_{\text{opt}}$ can be any standard HPO method (e.g., TPE, BOHB) that supports warm starts. All calls reuse the cumulative history $(\mathcal{H}, \mathcal{Y})$, enabling consistent importance estimation and avoiding redundant random initialization. In our experiments, we focus on the scheduling strategy itself and therefore adopt TPE as the default $\mathcal{A}_{\text{opt}}$ unless otherwise specified.

## 5 EXPERIMENTS

We evaluated GIF under fixed budgets on three classes of test cases: (1) anisotropic analytic functions designed to stress high-dimensional search, (2) Bayesmark tabular tasks with multiple models and datasets, and (3) NAS-Bench-301 (33D) neural architecture tuning. Unless otherwise stated, each run used a total budget of 500 evaluations across 5 independent seeds, with the initial warm-start budget $B_{\text{init}}{=}100$ counted toward the total budget. The optimizer $\mathcal{A}_{\text{opt}}$ was TPE (Akiba et al., 2019), and the importance evaluator $\mathcal{A}_{\text{imp}}$ was N-RReliefF (Wang et al., 2024). For GIF, we adopted a practical default configuration: sample ratio $\alpha{=}0.6$, step size $B_{\text{step}}{=}d$ evaluations per round, maximum group size $k{=}\lfloor d/3 \rfloor$, and a full-space fallback ratio $\rho{=}0.2$. These values were chosen as reasonable defaults for consistency across benchmarks, though other settings could also be applied in practice.

## 5.1 ANISOTROPIC ANALYTIC FUNCTION BENCHMARKS

We selected five classic black-box optimization functions widely used in HPO benchmarking: Sphere, Rosenbrock, Ackley (Ackley, 2012), Griewank (Griewank, 1985), and Rastrigin (Rastrigin, 1974). Each function was instantiated at dimensions $d \in \{5, 10, 30, 50\}$.

**Anisotropic Variable Transformation.** To induce anisotropy, we applied a diagonal scaling $\mathbf{w} = (w_1, \ldots, w_d)$ with $w_i = \exp(-\alpha(i-1))$, $\alpha = \frac{-\log(10^{-3})}{d-1}$, so that $w_d/w_1 = \exp(-\alpha(d-1)) = 10^{-3}$. This stylized construction creates a known, non-uniform sensitivity profile across coordinates. It is intended to provide a clean and controlled testbed for evaluating importance-aware schedulers under heterogeneous influence.

| Function | Formula |
|---|---|
| Anisotropic Sphere | $f(\mathbf{x}) = -\sum_{i=1}^{d}(w_i x_i)^2$ |
| Anisotropic Rosenbrock | $f(\mathbf{x}) = -\sum_{i=1}^{d-1}\left[100\big(w_{i+1}x_{i+1} - (w_i x_i)^2\big)^2 + (1 - w_i x_i)^2\right]$ |
| Anisotropic Ackley | $f(\mathbf{x}) = -\left(-20\exp\left(-0.2\sqrt{\frac{1}{d}\sum(w_i x_i)^2}\right) - \exp\left(\frac{1}{d}\sum\cos(2\pi w_i x_i)\right) + 20 + e\right)$ |
| Anisotropic Griewank | $f(\mathbf{x}) = -\left(1 + \frac{1}{4000}\sum(w_i x_i)^2 - \prod_{i=1}^{d}\cos\left(\frac{w_i x_i}{\sqrt{i}}\right)\right)$ |
| Anisotropic Rastrigin | $f(\mathbf{x}) = -\sum_{i=1}^{d}\left[(w_i x_i)^2 - 10\cos(2\pi w_i x_i) + 10\right]$ |

Table 1: **Weighted analytic benchmark functions with anisotropic scaling.** Domains: $[-5, 5]^d$ for Sphere, Rosenbrock, Ackley, and Griewank; $[-5.12, 5.12]^d$ for Rastrigin. We negate the standard minimization forms to adopt a maximization convention. For Sphere, Ackley, Griewank, and Rastrigin, the global maximizer is $\mathbf{x} = \mathbf{0}$ with maximum 0. For Rosenbrock, the *unconstrained* maximizer under our scaling satisfies $w_i x_i = 1$ for all $i$, yielding value 0.

**Baselines.** We compared GIF against Sequential Grouping (SG) (Wang et al., 2025), Bayesian Optimization based on Tree-structured Parzen Estimator (TPE), Bayesian Optimization based on Gaussian Process (GP), Bayesian Optimization with Hyperband (BOHB) (Falkner et al., 2018), and Random Search. All competitors used the identical box domains in Table 1, the same total evaluation budget (500) and seeds, and — where appropriate — the same warm-start history.

**Verification of Importance Estimation.** We verified that N-RReliefF could serve as an importance analyzer by testing whether it recovered the coordinate-wise anisotropy of each benchmark function. For every function and each $d \in \{5, 10, 30, 50\}$, we drew 500 i.i.d. samples $\mathbf{x} \sim \mathcal{U}([-1, 1]^d)$, evaluated $y = f(\mathbf{x})$, estimated per-coordinate importances $\{I_i\}$, and compared them with the ground-truth weights $\{w_i\}$ after max-normalization. Recovery was assessed by visually overlaying the two normalized curves and by reporting the Pearson correlation between $\{w_i\}$ and $\{I_i\}$.

### 5.1.1 ABLATION STUDIES

To isolate the contribution of each design component in GIF, we conducted ablations on the same anisotropic analytic benchmarks as Table 1, using the identical protocol and budgets as in the previous subsection. **Variant A — Randomized Importance (`RandImp`):** We replaced the importance evaluator with random per–coordinate weights to test whether gains arose from meaningful importance estimation rather than staged optimization alone; **Variant B — Uniform Allocation (`UniAlloc`):** We retained true importances for grouping but allocated an equal number of trials to each group (no importance weighting) to probe the necessity of importance-weighted budgeting; **Variant C — No Full-Space Fallback (`NoFB`):** We disabled the joint full-space optimization step to evaluate the fallback's role in escaping local plateaus and maintaining robustness.

For all the experiments on the anisotropic analytic benchmarks, we aggregated across all five functions and $d \in \{5, 10, 30, 50\}$, and reported: (i) best-objective convergence curves vs. evaluations (mean $\pm$ sem over 5 seeds; App. D); (ii) final best values at 500 trials summarized in a per-function heatmap (mean $\pm$ std across seeds; Fig. 11); and (iii) normalized regret AUC (lower is better; Table 3).

## 5.2 BAYESMARK

Bayesmark is an open-source benchmark for comparing Bayesian optimization methods via a unified API, standardized search spaces, and consistent evaluation (Turner & Eriksson, 2019). We ran

the official benchmark on four datasets (`breast`, `digits`, `iris`, `wine`) and *four models* (RF, DT, MLP-adam, MLP-sgd). Because GIF targets higher-dimensional HPO, we focused on the Bayesmark tasks with the highest dimensionalities within the suite ($d \approx 6$–10; see Table 6 for exact dimensionalities). For consistency across tasks, we use accuracy throughout. Train/validation splits and hyperparameter ranges followed Bayesmark defaults for all optimizers. We summarized performance via task-wise normalized final best scores (Perf. Norm), Avg. Rank, Time Rank, and Win Rate in Table 4 (see formulation in App. E.5). And App. E.4 provides a compact task-level table comparing GIF to the strongest baseline per dataset–model pair (mean accuracy over 5 seeds), along with *Score Gain* and *Time Saved*.

**Baselines.** We included Bayesmark's default optimizers: *HyperOpt*, *OpenTuner-BanditA*, *OpenTuner-GA*, *OpenTuner-GA-DE*, *PySOT*, *RandomSearch*, *Scikit-GBRT-Hedge*, *Scikit-GP-Hedge*, *Scikit-GP-LCB*. To ensure fairness, all baselines and *GIF* ran with identical search spaces, budgets, seeds, and splits; when applicable, we reused the same warm-start history. The descriptions of each baseline optimizer are provided in App. E.1.

### 5.3 NAS-BENCH-301

Unlike the fully tabular NAS benchmarks 101 and 201 (Ying et al., 2019; Dong & Yang, 2020), NAS-Bench-301 (NB301)(Zela et al., 2020) is a surrogate benchmark that emulates the Differentiable Architecture Search (DARTS) (Liu et al., 2018) search space and yields fast, approximate evaluations in a realistic high-dimensional regime. Concretely, NB301 is built on the DARTS cell space trained on CIFAR-10 and provides learned regressors that map an architecture encoding to predicted validation accuracy (and a separate regressor for runtime), enabling faithful anytime comparisons without re-training each architecture. In this work, we used the official SNB-DARTS-XGB-v1.0 release (Zela et al., 2020): an XGBoost-based surrogate trained on DARTS+CIFAR-10 with stratified train/val/test splits over data gathered from multiple NAS optimizers. We kept the benchmark's 33-dimensional architecture encoding and queried the surrogate-predicted validation accuracy as the objective; for wall-clock plots we used the benchmark's runtime surrogate to accumulate simulated time. We evaluated GIF against TPE, BOHB, Random, and SG on darts-xgb-v1.0. We reported: (i) best validation score vs. evaluations; (ii) best validation score vs. simulated wall-clock time; and (iii) a Pareto view (score vs. time) that summarizes the quality–time trade-off (Fig. 5).

## 6 RESULTS

### 6.1 VERIFICATION OF IMPORTANCE ESTIMATION

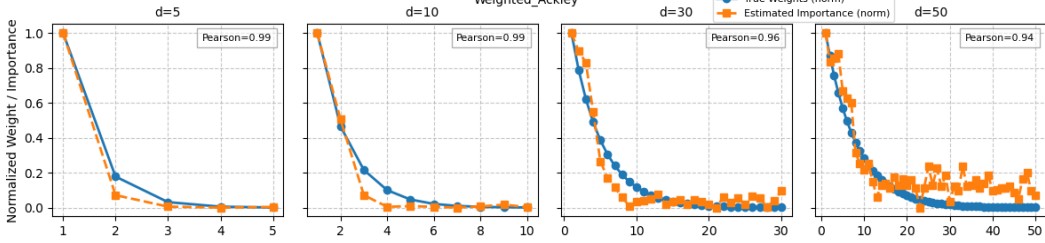

Figure 2: Weighted Ackley: ground truth weights $w_i$ vs. estimated weights $I_i$

Before applying GIF to real HPO tasks, we verify that N-RReliefF produces hyperparameter-importance scores $\{I_i\}$ that align with the generating anisotropy weights $\{w_i\}$ on weighted analytic benchmarks. Table 2 reports Pearson correlations between $\{I_i\}$ and $\{w_i\}$ across dimensions $d \in \{5, 10, 30, 50\}$: alignment is very strong in low dimensions (mean

| | $d = 5$ | $d = 10$ | $d = 30$ | $d = 50$ |
|---|---|---|---|---|
| Weighted Ackley | 0.995 | 0.986 | 0.959 | 0.941 |
| Weighted Griewank | 0.985 | 0.896 | 0.670 | 0.547 |
| Weighted Rastrigin | 0.990 | 0.854 | 0.696 | 0.717 |
| Weighted Rosenbrock | 0.993 | 0.982 | 0.831 | 0.791 |
| Weighted Sphere | 0.987 | 0.917 | 0.819 | 0.805 |
| Mean ± Std | 0.990 ± 0.004 | 0.927 ± 0.057 | 0.795 ± 0.116 | 0.760 ± 0.144 |

Table 2: Pearson correlation between ground-truth weights $w_i$ and HIA score estimates $I_i$.

$r \approx 0.95$–$0.99$). And as the dimensionality grows, the effective data per dimension shrinks under a fixed budget, leading to noisier importance estimates and hence lower correlations. But they still remain consistently above 0.7 on average. As a representative illustration, Fig. 2 overlays $\{w_i\}$ and $\{I_i\}$ for the weighted Ackley function; it shows the best case in terms of Pearson's linear correlation (near-perfect at $d = 5, 10$ with $r \approx 0.99$, and only slight drops at $d = 30, 50$ with $r = 0.96, 0.94$). Other benchmarks also exhibit similar monotone relationships between $\{I_i\}$ and $\{w_i\}$, though their Pearson correlations are not as high as in the Ackley case. See App. C for the full grid of plots.

## 6.2 ANALYTIC BENCHMARKS AND ABLATIONS

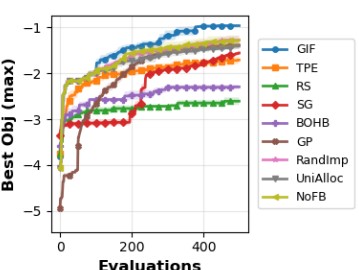

Figure 3: Convergence on Weighted Ackley (50D).

We benchmark GIF on five anisotropic analytic functions at $d \in \{5, 10, 30, 50\}$ against TPE, BOHB, GP, RS, and SG (protocol in Sec. 5.1, ablation setup in Sec. 5.1.1). Under the normalized regret AUC metric (see App. D.3), GIF is comparable to TPE at $d{=}5$ (small spaces yield limited benefit from importance-aware scheduling) and becomes consistently stronger for $d{\geq}10$: relative to the best baseline it reduces regret AUC by **35%** at $d{=}10$ (vs. BOHB), **31%** at $d{=}30$ (vs. TPE), and **33%** at $d{=}50$ (vs. GP). Fig. 3 illustrates a typical 50D case (Weighted Ackley), where GIF maintains a consistently higher best-so-far curve at matched evaluation counts.

Ablations in Table 3 show each component matters: replacing learned importance with random weights (RandImp) or using uniform per-group allocation (UniAlloc) degrades AUC, confirming the value of (i) stable importance ordering and (ii) proportional budgeting; removing the full-space fallback (NoFB) hurts most in high-$d$, evidencing the need for periodic global exploration to escape misleading subspaces. Per-function convergence and variability heatmaps are in App. D; on Rosenbrock the margin narrows due to strong inter-variable coupling, but GIF remains competitive overall.

| Dim | BOHB | GP | RS | SG | TPE | GIF-win |
|---|---|---|---|---|---|---|
| 5 | $4.90 \pm 2.07$ | $10.40 \pm 3.27$ | $7.01 \pm 2.55$ | $9.57 \pm 1.36$ | $2.71 \pm 0.64$ | 20% |
| 10 | $11.36 \pm 3.16$ | $20.35 \pm 6.23$ | $21.41 \pm 2.72$ | $25.36 \pm 5.01$ | $8.06 \pm 2.00$ | 60% |
| 30 | $35.09 \pm 3.42$ | $30.89 \pm 4.52$ | $39.75 \pm 6.01$ | $40.14 \pm 5.42$ | $21.34 \pm 2.84$ | 100% |
| 50 | $44.23 \pm 3.57$ | $42.88 \pm 2.26$ | $48.36 \pm 4.28$ | $47.08 \pm 6.15$ | $29.35 \pm 2.54$ | 100% |

| Dim | GIF | RandImp | UniAlloc | NoFB | GIF-win |
|---|---|---|---|---|---|
| 5 | $3.20 \pm 0.81$ | $3.24 \pm 2.65$ | $3.28 \pm 2.63$ | $3.26 \pm 2.68$ | 0% |
| 10 | $\mathbf{7.36 \pm 0.96}$ | $6.96 \pm 2.55$ | $7.95 \pm 2.68$ | $7.19 \pm 2.51$ | 60% |
| 30 | $\mathbf{14.75 \pm 2.55}$ | $18.46 \pm 3.98$ | $21.01 \pm 5.71$ | $17.62 \pm 4.39$ | 100% |
| 50 | $\mathbf{19.18 \pm 1.56}$ | $24.07 \pm 3.07$ | $24.70 \pm 2.97$ | $23.33 \pm 2.54$ | 100% |

Table 3: Normalized regret AUC (lower is better) for anisotropic analytic benchmarks. Top: baselines vs. GIF. Bottom: ablations. GIF-win = fraction of seeds with the best AUC.

| Opt. | Avg. Rank ↓ | Time Rank ↓ | Perf. Norm ↑ | Win Rate ↑ |
|---|---|---|---|---|
| GIF | **2.72** | 5.13 | **0.811** | **0.750** |
| RS | 6.28 | **1.66** | 0.189 | 0.063 |
| HOpt | 4.56 | 5.84 | 0.354 | 0.094 |
| PySOT | 4.47 | 6.03 | 0.371 | 0.156 |
| GP-H | 3.47 | 10.0 | 0.401 | 0.125 |
| OT-B | 5.28 | 3.78 | 0.274 | 0.094 |
| GBRT | 4.78 | 7.94 | 0.314 | 0.125 |
| OT-GD | 5.50 | 2.94 | 0.234 | 0.063 |
| GP-LCB | 4.09 | 9.00 | 0.370 | 0.219 |
| OT-GA | 5.75 | 2.69 | 0.220 | 0.063 |

Table 4: Bayesmark summary. Optimizer abbreviations follow App. E.1.

## 6.3 BAYESMARK (MID-DIMENSIONAL EVALUATION)

Having established GIF's behavior on anisotropic analytic functions, we next turn to a mid-dimensional, real-world suite: *Bayesmark*. Table 4 reports several aggregate metrics whose detailed definitions are given in App. E.5. Briefly, (i) Perf. Norm = mean normalized final score across tasks (per-task scores are min–max normalized after seed averaging); (ii) Avg. Rank and Time Rank = average rank across tasks by performance and runtime; (iii) Win Rate = fraction of task–seed instances where an optimizer achieves the top score. Under these definitions, GIF attains the

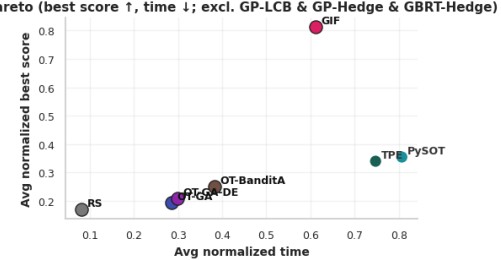

Figure 4: Pareto trade-off between final score and time (lightweight methods).

highest normalized performance (0.811) and win rate (0.750), with the best average rank (2.72). In

Fig. 4, we omit GP-LCB, GP-Hedge, and GBRT-Hedge since their wall-clock times are dominated by the cost of repeatedly training large surrogate models. GP-LCB and GP-Hedge update a full Gaussian process after every trial, which scales cubically in the number of observations and quickly becomes prohibitive. GBRT-Hedge trains ensembles of regression trees at each iteration, which similarly incurs non-negligible overhead. Including them would push the x-axis far to the right, squashing the remaining points into a narrow cluster and obscuring the score–time trade-offs. Consequently, the figure focuses on lightweight methods to reveal the Pareto structure. Taken together, the Bayesmark experiments confirm that GIF consistently converts its evaluation efficiency into higher normalized performance and win rate, while maintaining relatively competitive wall-clock times.

## 6.4 NASBENCH 301 (HIGH-DIMENSIONAL EVALUATION)

We now shift to a genuinely high-dimensional setting: NAS-Bench-301 (33D, DARTS-XGB surrogate). Figure 5 summarizes convergence in evaluations (left), wall-clock time (middle), and the score–time Pareto view (right). In evaluations, GIF keeps improving after other methods flatten out; around 340 evaluations, it overtakes all baselines. This indicates that the importance-guided scheduler continues to discover productive subspaces late in the run, and the warm-started full-space fallback helps it escape plateaus. In wall-clock time, GIF uses the budget efficiently: it reaches the top accuracy without being the slowest; SG is faster but stalls at a lower ceiling, and GP is slowest at the same budget. The Pareto panel makes the trade-off explicit: GIF and SG define the frontier—SG at the "faster but lower score" end, GIF at the "higher score at similar time" end—while GP, TPE, BOHB, and Random are dominated (either slower for similar accuracy or less accurate at similar time). As a result, in high dimensions, focusing trials on the most important groups while retaining periodic full-space search yields stronger final incumbents and higher accuracy for the time spent.

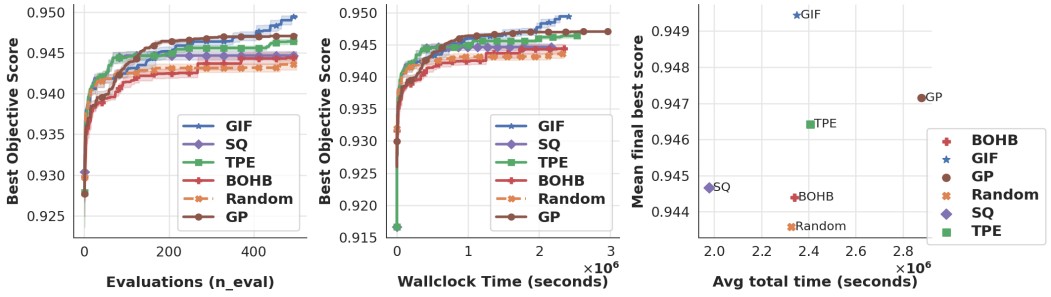

Figure 5: Convergence and Pareto analysis on NAS-Bench-301 (DARTS-XGB surrogate, 33D)

## 7 CONCLUSION

Our study introduces GREEDY IMPORTANCE FIRST (GIF), an importance-aware strategy that translates early hyperparameter-importance estimates into concrete scheduling decisions—grouping by importance, proportional allocation, and a safeguarded full-space fallback. Across diverse benchmarks, a consistent pattern emerges: GIF is most effective in high-dimensional regimes. On weighted analytic functions and NAS-Bench-301, it achieves both faster convergence and stronger final incumbents than strong baselines. On Bayesmark, where the effective dimensionality is smaller, GIF remains competitive, but its margins are limited on simpler models and become most pronounced on the MLPs—reflecting that importance-guided scheduling yields the biggest gains when many low-impact variables dilute progress and the landscape exhibits stronger anisotropy. In summary, GIF provides a simple, plug-compatible pathway to sample-efficient HPO in high dimensions; by reweighting effort toward important subspaces while maintaining a robust fallback, it offers practical utility for deep learning model tuning, and lays a foundation for future research on importance-aware AutoML systems.

## ETHICS STATEMENT

We affirm adherence to the ICLR Code of Ethics. We did not collect new data, use human subjects, or process personally identifiable information. All datasets and benchmarks are publicly available under their respective licenses, and we followed their usage guidelines. The method is optimizer-agnostic and could be applied to high-stakes domains; in such cases, deployers should conduct problem-specific safety, privacy, and fairness assessments (e.g., dataset documentation, bias audits, and model risk evaluation) before use. The authors are unaware of any conflicts of interest that would unduly influence the reported results.

## REPRODUCIBILITY STATEMENT

We took several steps to enable reproducibility. Algorithmic components and scheduling logic are specified in Secs. 4.1–4.6, with pseudocode for all routines in App. B (Algorithms 1–6). The importance estimator (N-RReliefF) and its normalization are detailed in App. A (Alg. 2). Experimental settings—including budgets, seeds, default GIF hyperparameters ($\alpha$=0.6, $B_{\text{step}}$=d, $k$=$\lfloor d/3 \rfloor$, $\rho$=0.2), and optimizer choices (TPE for $\mathcal{A}_{\text{opt}}$)—are described in Sec. 5. The anisotropic analytic functions, domains, and transformations are provided in Sec. 5.1 and Table 1. Bayesmark tasks, datasets, and model search spaces are specified in Sec. 6.3 and Apps. E.2–E.3, with baseline descriptions in App. E.1. NAS-Bench-301 details (the SNB-DARTS-XGB-v1.0 surrogate) appear in Sec. 6. Our primary metric (normalized regret AUC) is precisely defined in App. D.3; verification plots for importance recovery are in App. C and additional convergence summaries in App. D. We will include an anonymized code archive in the supplementary materials with configuration files, fixed random seeds, and scripts to regenerate tables and figures, as well as raw logs for the reported runs, to facilitate independent reproduction. Anonymous repository is available at https://anonymous.4open.science/r/ICLR_GIF-F175/.

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

## A  DETAILS OF HIA (N-RRELIEFF)

As a lineage, N-RReliefF (Wang et al., 2024) descends from the original ReliefF feature-weighting algorithm and its regression/continuous-output extensions (Kononenko, 1994; Robnik-Šikonja & Kononenko, 2003). We instantiate HIA with *N-RReliefF*, treating each tried configuration $\mathrm{h}^{(r)} = (h_1^{(r)}, \ldots, h_d^{(r)})$ as an instance and its score $p^{(r)}$ as the target. A mixed distance between two configurations $\mathrm{h}, \mathrm{h}'$ is computed as

$$\mathrm{dist}(\mathrm{h}, \mathrm{h}') = \sum_{i=1}^{d} \mathrm{diff}_i(h_i, h_i'), \quad \mathrm{diff}_i(h_i, h_i') = \begin{cases} \dfrac{|h_i - h_i'|}{\mathrm{range}(\Theta_i)} & \text{if } H_i \text{ is continuous,} \\ \mathbb{1}[h_i \neq h_i'] & \text{if } H_i \text{ is categorical,} \\ 0 & \text{if } H_i \text{ is inactive (conditional).} \end{cases}$$

Performance values are min–max normalized on the set of observed performance values $\mathcal{P}$ to $[0, 1]$ before use. Let $n_{\mathrm{ref}}$ be the number of reference trials and $n_{\mathrm{nbr}}$ the number of nearest neighbors per reference (in configuration space). For each reference $r$ and its neighbor set $\mathcal{N}(r)$ with $|\mathcal{N}(r)| = n_{\mathrm{nbr}}$, we accumulate per-dimension covariation:

$$\widehat{I}_i = \frac{1}{n_{\mathrm{ref}} n_{\mathrm{nbr}}} \sum_{r=1}^{n_{\mathrm{ref}}} \sum_{n \in \mathcal{N}(r)} \mathrm{diff}_i\big(h_i^{(r)}, h_i^{(n)}\big) \phi\big(p^{(r)}, p^{(n)}\big), \quad \phi(p, p') = |p - p'|.$$

Finally, we map raw scores to positive, comparable importances via temperature-controlled softplus normalization:

$$\bar{I} = \frac{1}{d} \sum_{j=1}^{d} \widehat{I}_j, \quad s_\tau(z) = \tau \log(1 + e^{z/\tau}), \ \tau > 0, \quad I_i = \frac{s_\tau(\widehat{I}_i - \bar{I})}{\sum_{j=1}^{d} s_\tau(\widehat{I}_j - \bar{I})}.$$

By default we set $n_{\text{ref}} = \min\{200, t\}$, where $t = |\mathcal{H}|$ is the current number of evaluated trials, and use $n_{\text{nbr}} = 10$ neighbors. Pairwise scores $\widehat{I}_{i,j}$ for analysis can be obtained by replacing $\text{diff}_i(\cdot)$ with $\text{diff}_i(\cdot)\text{diff}_j(\cdot)$ and normalizing analogously. Unless otherwise stated, GIF uses only $\{I_i\}$ for grouped scheduling.

---

**Algorithm 2** N-RReliefF for Hyperparameter Importance

---

**Require:** Evaluated trials $D = \{(h^{(r)}, p^{(r)})\}_{r=1}^t$, number of references $n_{\text{ref}}$, number of neighbors $n_{\text{nbr}}$

**Ensure:** Normalized importance scores $\{I_i\}_{i=1}^d$ (and optionally pairwise scores $\{I_{i,j}\}$)

1: Initialize raw scores $\widehat{I}_i \leftarrow 0$ for all $i = 1, \ldots, d$
2: **for** $r = 1$ to $n_{\text{ref}}$ **do**
3:     Sample reference $(h^{(r)}, p^{(r)})$ from $D$
4:     Find $n_{\text{nbr}}$ nearest neighbors $\mathcal{N}(r)$ of $h^{(r)}$
5:     **for all** $n \in \mathcal{N}(r)$ **do**
6:       **for** $i = 1$ to $d$ **do**
7:         $\widehat{I}_i \leftarrow \widehat{I}_i + \text{diff}_i(h_i^{(r)}, h_i^{(n)}) \cdot |p^{(r)} - p^{(n)}|$
8:         Optionally: $\widehat{I}_{i,j} \leftarrow \widehat{I}_{i,j} + \text{diff}_i(\cdot)\,\text{diff}_j(\cdot)\,|p^{(r)} - p^{(n)}|$
9:       **end for**
10:     **end for**
11: **end for**
12: Normalize $\widehat{I}_i \leftarrow \frac{1}{n_{\text{ref}} n_{\text{nbr}}} \widehat{I}_i$ for each $i$
13: Compute $\bar{I} = \frac{1}{d} \sum_{j=1}^d \widehat{I}_j$
14: **for** $i = 1$ to $d$ **do**
15:     $I_i \leftarrow \dfrac{s_\tau(\widehat{I}_i - \bar{I})}{\sum_{j=1}^d s_\tau(\widehat{I}_j - \bar{I})}$ with $s_\tau(z) = \tau \log(1 + e^{z/\tau})$
16: **end for**
17: **return** $\{I_i\}$ (and optionally $\{I_{i,j}\}$)

---

## B  PSEUDOCODE OF GIF KEY COMPONENTS

---

**Algorithm 3** warm-start

---

**Require:** Search space $\Theta$, objective $f_{\mathcal{D}}$, subsample ratio $\alpha \in (0, 1]$, init budget $B_{\text{init}}$, optimizer $\mathcal{A}_{\text{opt}}$

**Ensure:** Initial history $(\mathcal{H}, \mathcal{Y})$, incumbent $(\mathbf{h}_{\text{best}}, y_{\text{best}})$

1: $\mathcal{D}_{\text{init}} \leftarrow$ randomly subsample $\mathcal{D}$ by ratio $\alpha$
2: $(\mathcal{H}, \mathcal{Y}) \leftarrow \mathcal{A}_{\text{opt}}(f_{\mathcal{D}}, \Theta, \mathcal{D}_{\text{init}}, B_{\text{init}})$
3: $(\mathbf{h}_{\text{best}}, y_{\text{best}}) \leftarrow \arg\max_{(h,y) \in (\mathcal{H}, \mathcal{Y})} y$
4: **return** $(\mathcal{H}, \mathcal{Y}), (\mathbf{h}_{\text{best}}, y_{\text{best}})$

---

---

**Algorithm 4** AllocateBudget

---

**Require:** Groups $\mathcal{G} = \{\mathcal{G}_1, \ldots, \mathcal{G}_m\}$, per-HP importances $\{I_i\}_{i=1}^d$, round budget $B$
**Ensure:** Group allocations $\mathbf{b} = [b_1, \ldots, b_m]$, $\sum_j b_j \leq B$

1: $I_{\text{total}} \leftarrow \sum_{i=1}^d I_i$
2: **for** $j = 1$ **to** $m$ **do**
3:    $I_j \leftarrow \sum_{i \in \mathcal{G}_j} I_i$
4:    $b_j \leftarrow \left\lfloor \frac{I_j}{I_{\text{total}}} B \right\rfloor$
5:    $b_j \leftarrow \max(1, b_j)$ {ensure nonzero coverage; can be disabled if skipping is allowed}
6: **end for**
7: $S \leftarrow \sum_{j=1}^m b_j, \quad d \leftarrow B - S$
8: **if** $d > 0$ **then**
9:    $t \leftarrow \arg\max_{1 \leq j \leq m} I_j$ {give remainder to most important group}
10:    $b_t \leftarrow b_t + d$
11: **end if**
12: **return** $\mathbf{b}$

---

**Algorithm 5** GroupOptimization

---

**Require:** Groups $\mathcal{G} = \{\mathcal{G}_j\}$, allocations $\mathbf{b} = [b_1, \ldots, b_{|\mathcal{G}|}]$, history $(\mathcal{H}, \mathcal{Y})$,
    incumbent $(\mathbf{h}_{\text{best}}, y_{\text{best}})$, current trials $T_{\text{used}}$, optimizer $\mathcal{A}_{\text{opt}}$, total budget $B_{\text{total}}$
**Ensure:** Updated $(\mathcal{H}, \mathcal{Y})$, $(\mathbf{h}_{\text{best}}, y_{\text{best}})$, $T_{\text{used}}$, flag *improved*

1: *improved* $\leftarrow$ False
2: **for** $j = 1$ **to** $|\mathcal{G}|$ **do**
3:    Fix all hyperparameters outside $\mathcal{G}_j$ to their values in $h_{\text{best}}$
4:    $(\mathcal{H}_{\text{new}}, \mathcal{Y}_{\text{new}}) \leftarrow \mathcal{A}_{\text{opt}}(f_{\mathcal{D}}, \mathcal{G}_j, b_j, \text{fixed} = h_{\text{best}-\mathcal{G}_j}, \text{warm} - \text{start} = (\mathcal{H}, \mathcal{Y}))$
5:    $\mathcal{H} \leftarrow \mathcal{H} \cup \mathcal{H}_{\text{new}}, \quad \mathcal{Y} \leftarrow \mathcal{Y} \cup \mathcal{Y}_{\text{new}}$ {append to history if order matters}
6:    $T_{\text{used}} \leftarrow T_{\text{used}} + b_j$
7:    $(h^*, y^*) \leftarrow \arg\max_{(h,y) \in (\mathcal{H}, \mathcal{Y})} y$
8:    **if** $y^* > y_{\text{best}}$ **then**
9:       $(\mathbf{h}_{\text{best}}, y_{\text{best}}) \leftarrow (h^*, y^*); \quad$ *improved* $\leftarrow$ True
10:    **end if**
11:    **if** $T_{\text{used}} \geq B_{\text{total}}$ **then**
12:       **break**
13:    **end if**
14: **end for**
15: **return** $(\mathcal{H}, \mathcal{Y})$, $(\mathbf{h}_{\text{best}}, y_{\text{best}})$, $T_{\text{used}}$, *improved*

---

**Algorithm 6** FullSpaceOptimization

---

**Require:** Search space $\Theta$, objective $f_{\mathcal{D}}$, budget $B_{\text{full}}$, history $(\mathcal{H}, \mathcal{Y})$, incumbent $(\mathbf{h}_{\text{best}}, y_{\text{best}})$,
    current trials $T_{\text{used}}$, used fallback quota $T_{\text{full used}}$, optimizer $\mathcal{A}_{\text{opt}}$
**Ensure:** Updated $(\mathcal{H}, \mathcal{Y})$, $(\mathbf{h}_{\text{best}}, y_{\text{best}})$, $T_{\text{used}}$, $T_{\text{full used}}$

1: $(\mathcal{H}_{\text{full}}, \mathcal{Y}_{\text{full}}) \leftarrow \mathcal{A}_{\text{opt}}(f_{\mathcal{D}}, \Theta, B_{\text{full}}, \text{warm} - \text{start} = (\mathcal{H}, \mathcal{Y}))$
2: $\mathcal{H} \leftarrow \mathcal{H} \cup \mathcal{H}_{\text{full}}, \quad \mathcal{Y} \leftarrow \mathcal{Y} \cup \mathcal{Y}_{\text{full}}$
3: $T_{\text{used}} \leftarrow T_{\text{used}} + B_{\text{full}}; \quad T_{\text{full used}} \leftarrow T_{\text{full used}} + B_{\text{full}}$
4: $(h^*, y^*) \leftarrow \arg\max_{(h,y) \in (\mathcal{H}, \mathcal{Y})} y$
5: **if** $y^* > y_{\text{best}}$ **then**
6:    $(\mathbf{h}_{\text{best}}, y_{\text{best}}) \leftarrow (h^*, y^*)$
7: **end if**
8: **return** $(\mathcal{H}, \mathcal{Y})$, $(\mathbf{h}_{\text{best}}, y_{\text{best}})$, $T_{\text{used}}$, $T_{\text{full used}}$

---

## C PEARSON CORRELATION VERIFICATION OF ANISOTROPY ON ALL ANISOTROPIC ANALYTIC FUNCTIONS

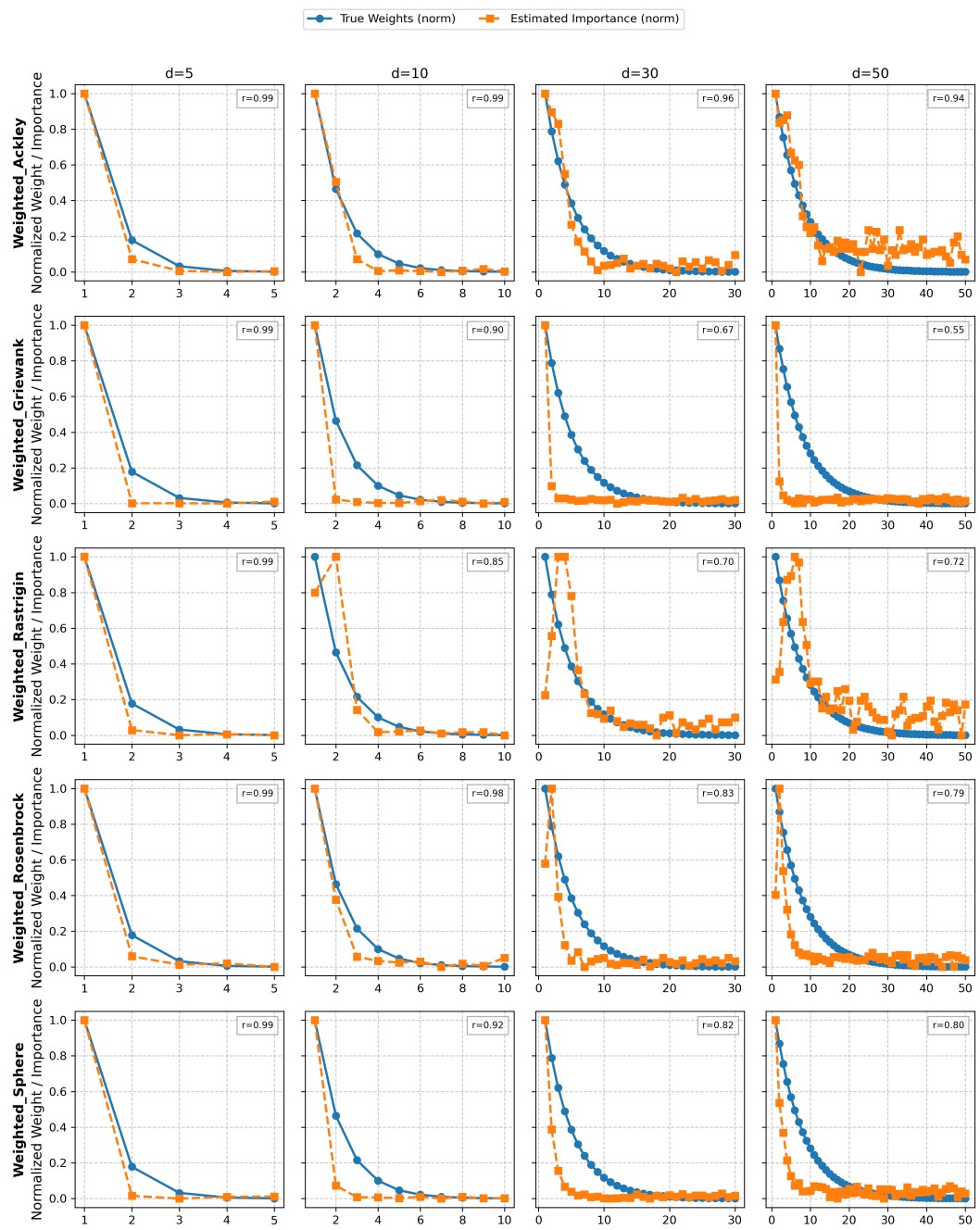

Figure 6: **Anisotropy verification across five anisotropic analytic benchmark functions.** Rows are functions and columns are $d \in \{5, 10, 30, 50\}$. Blue: normalized ground-truth weights; orange: normalized N-RReliefF estimates; panels show Pearson $r$.

Across all functions, N-RReliefF tracks the overall decay pattern of the true weights well in low dimensions ($d = 5, 10$), yielding high correlations ($r \gtrsim 0.9$). As $d$ increases ($d = 30, 50$), correlations decrease due to high-dimensional sparsity, which weakens nearest-neighbor estimates that N-RReliefF relies on. Importantly, despite this drop in $r$, the estimated importance curves still

clearly preserve the expected monotonic decay: large-weight coordinates remain dominant and small-weight coordinates remain negligible, so the anisotropy structure is still recoverable. Among functions, Weighted Ackley remains the most robust (e.g., $r \approx 0.94$ at $d = 50$), Weighted Sphere is also stable thanks to its convex, single-peaked landscape; Weighted Rosenbrock degrades moderately due to curved valleys and cross-dimensional dependencies; Weighted Rastrigin drops more with $d$ because multimodality injects noise into local neighborhoods; and Weighted Griewank is the most challenging in high dimensions due to oscillatory product terms that induce nonlocal interactions. Overall, these results indicate that (i) anisotropy is accurately captured in small/medium $d$, (ii) robustness depends on landscape complexity, and (iii) even when $r$ declines at large $d$, the qualitative decay trend of importance is retained.

# D ADDITIONAL RESULTS FOR ANALYTIC BENCHMARKS AND ABLATIONS

Figures 7–10 plot best objective (maximization) vs. evaluations for each of the five weighted analytic functions under $d=5, 10, 30, 50$ (symlog $y$-axis; shaded areas are $\pm$ SEM over seeds). Figure 11 summarizes, per function and method, the rank (top text) and the final best mean across seeds (bottom text), with color encoding std across seeds (darker = higher variability). Across dimensions, GIF generally improves faster and reaches stronger final incumbents than baselines, with variability comparable to or lower than alternatives, especially for $d \geq 30$.

## D.1 WHY GIF IS LESS DOMINANT AT LOW DIMENSIONS.

We observe that at $d=5$–10 (Figures 7, 8), classical model-based optimizers (e.g., TPE) often close the gap to GIF on most functions. This is expected for three reasons: (i) **Diminished benefit of importance estimation.** With few dimensions, anisotropy is easier to explore directly; the gain from HIA and forming importance-sorted groups is small because the search already covers the full space frequently. (ii) **Overhead vs. budget.** GIF spends part of the per-round budget on warm-start sampling, HIA (N-RReliefF), grouping and a full-space fallback. Under a small effective dimensionality, this coordination overhead yields less net advantage than in high-$d$ where focusing evaluations pays off. (iii) **Noise in early HIA.** Early HIA could be noisier, occasionally suggesting near-ties among variables or even failing to correctly estimate their relative weights. In such cases, TPE's direct modeling of the full space can perform better, which matches the tighter ranks and smaller GIF win rate at $d=5$–10 in Table 3.

## D.2 WHY THE MARGIN ON ROSENBROCK IS SMALLER AT HIGH DIMENSIONS.

On the weighted Rosenbrock, even for $d=30$–50 (Figures 9, 10), GIF's lead is narrower than on Ackley/Griewank/Rastrigin. Mechanistically: Rosenbrock features a narrow, curved valley with high parameter interactions. Progress requires coordinated multi-dim moves along the ridge. GIF's grouping improves sample allocation but, by construction, emphasizes variables by marginal importance; when interactions dominate, importance ordering is less separable and the benefit of importance-sorted, proportionally allocated subspaces is reduced.

**Takeaways.** (1) In low-$d$, GIF's coordination overhead and weaker need for importance-guided focus reduce its advantage; strong baselines already explore sufficiently. (2) In high-$d$, anisotropy and sparsity of useful directions make importance sorting + proportional allocation + fallback synergistic, producing larger gains and stable seeds. (3) On interaction-heavy landscapes like Rosenbrock, margins shrink because success hinges on coupled updates that are only partially captured by marginal-importance grouping; nevertheless, GIF remains competitive and typically top-ranked with reduced variability.

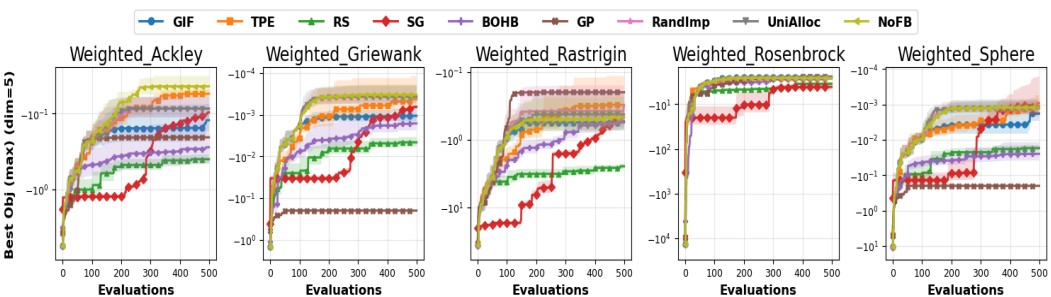

Figure 7: Best objective vs. evaluations on weighted analytic functions ($d$=5).

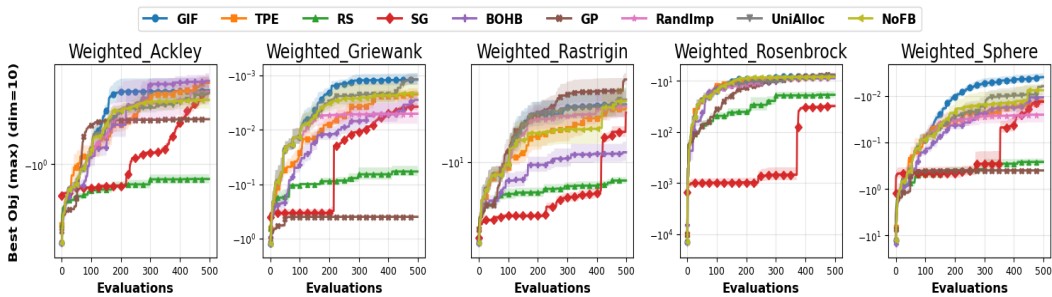

Figure 8: Best objective vs. evaluations on weighted analytic functions ($d$=10).

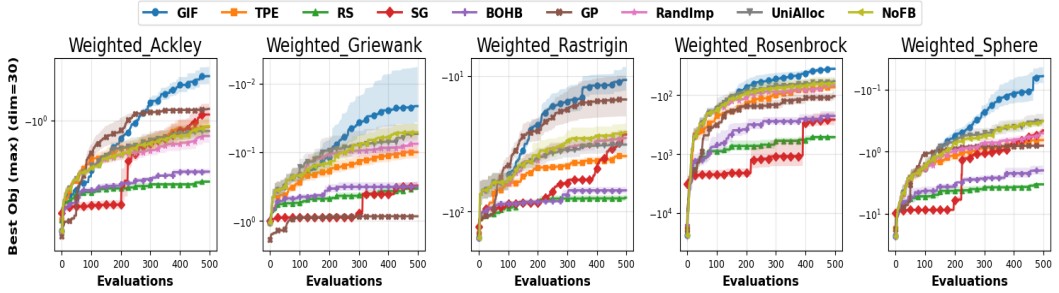

Figure 9: Best objective vs. evaluations on weighted analytic functions ($d$=30).

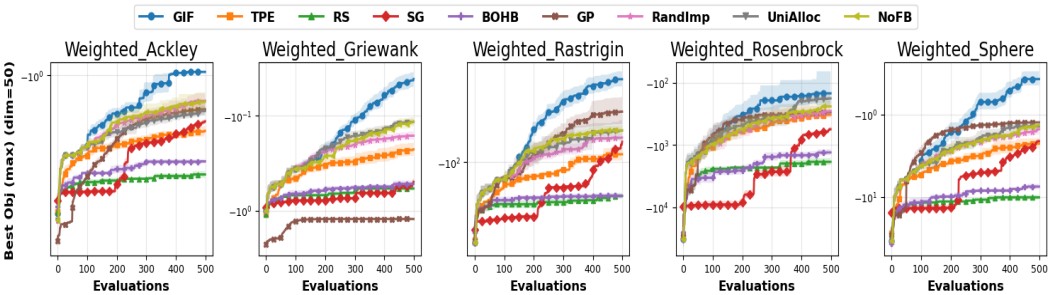

Figure 10: Best objective vs. evaluations on weighted analytic functions ($d$=50).

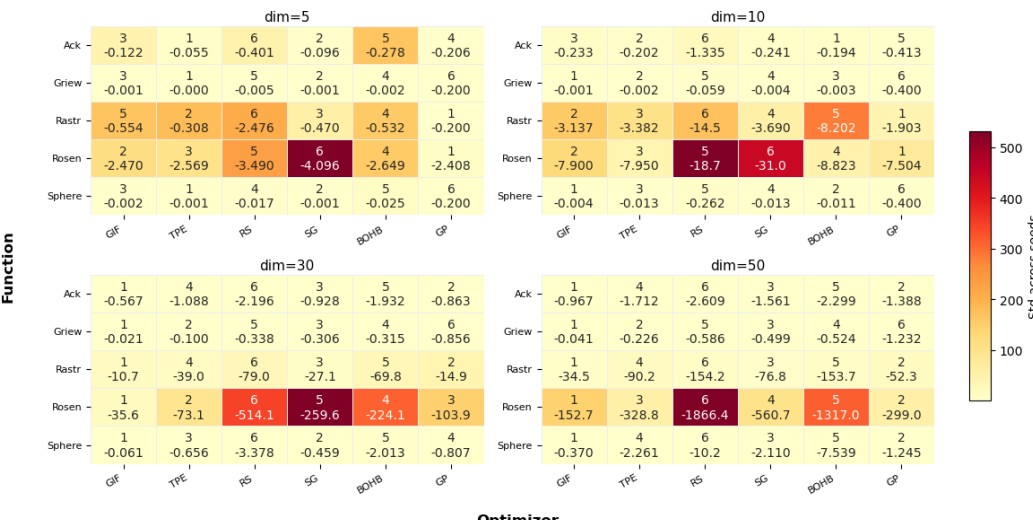

Figure 11: Performance summary of GIF and baselines on weighted analytic benchmarks.

Figure 11 indicates that at low dimensions ($d$=5), GIF's advantage is less pronounced (TPE and GP occasionally outperforms GIF), but as $d$ grows, GIF maintains strong incumbents while several baselines degrade or become unstable, especially on challenging functions such as Rosenbrock.

### D.3 EVALUATION METRIC: AGGREGATED REGRET AUC

In table 3, the score is assessed using the normalized regret area under the curve (regret AUC). For each trial $t$, regret is defined as

$$r_t = f^\star - \max_{s \leq t} f(h^{(s)}),$$

where $f^\star$ denotes a known upper bound of the maximization form (0 for our weighted analytic suite. The regret trajectory is then summarized by

$$\text{Regret-AUC} = \frac{1}{r_0 \cdot T} \int_0^T r_t \, dt,$$

where $T$ is the evaluation budget. For each $(f, dim, \text{seed})$, the initial regret $r_0$ is computed at trial 0, with the mean objective score across functions to provide a common baseline. Then, for each method, the regret trajectory on that function is integrated and divided by $r_0 \cdot T$, yielding a normalized regret AUC. Finally, normalized AUCs are averaged across the five functions at each dimension. This procedure eliminates scale discrepancies between functions (e.g., Sphere vs. Rastrigin), ensuring that no single function dominates the aggregated regret AUC due to its numerical range.

This aggregated regret AUC has been used in prior HPO work for comparing optimizers across heterogeneous tasks (Klein et al., 2017).

## E    ADDITIONAL DETAILS FOR BAYESMARK

### E.1    THE USED BASELINE OPTIMIZERS

- *HyperOpt (HOpt)*: Tree-structured Parzen Estimator (TPE) Bayesian optimization; it models $p(x \mid y)$ with Parzen density estimators and chooses $x$ to maximize the density ratio $l(x)/g(x)$ where $l$ and $g$ split observations by a quantile of $y$ (Bergstra et al., 2011). TPE handles mixed continuous/categorical spaces and conditionals natively and is robust with small initial designs. Key knobs: `algo` (`tpe.suggest`/`rand.suggest`/`atpe.suggest`), $\gamma$ (good–bad split, e.g., 0.15–0.25), KDE bandwidths, `n_startup_jobs`. Strengths: strong any-time performance;

scales to high-/discrete-dimensional spaces. Limitations: density estimation can degrade with heavily multi-modal/noisy objectives.

- *OpenTuner-BanditA (OT-B)*: OpenTuner's technique portfolio coordinated by a multi-armed bandit that allocates budget to competing search operators based on observed payoffs (Ansel et al., 2014). Operators include Nelder–Mead, PSO-like moves, GA-style mutations, and local search; BanditA adapts selection online. Pros: very adaptive across heterogeneous problems; no strict surrogate assumptions. Cons: more evaluations needed to identify winning operators; reproducibility depends on operator mix and seeds.
- *OpenTuner-GA (OT-GA)*: OpenTuner instantiated with genetic algorithm operators (selection/crossover/mutation) as primary movers (Ansel et al., 2014). Typical settings: population size, tournament size, crossover/mutation rates, and elitism; supports categorical/mixed spaces naturally. Works well when useful schemata exist or good configurations are recombinable; may struggle on deceptive/weakly heritable landscapes without auxiliary local search.
- *OpenTuner-GA-DE (OT-GD)*: Hybridizing genetic operators with differential evolution (DE) steps inside OpenTuner (Ansel et al., 2014). DE brings vector-based proposals (e.g., `DE/rand/1/bin`) helpful for continuous subspaces; GA maintains diversity for categorical parts. Pros: good balance of exploration/exploitation in mixed spaces. Cons: extra hyperparameters (F, CR, strategy) and interactions to tune.
- *PySOT (PySOT)*: Surrogate-assisted global optimization using RBF/GP/POU surrogates with adaptive sampling such as DYCORS and expected improvement tailored for expensive black boxes (Eriksson et al., 2019a). Supports parallel suggestion and trust-region style safeguards; good for smooth objectives under tight budgets. Pros: sample efficiency and principled infill; Cons: categorical handling requires encoding; performance depends on surrogate fit and scaling.
- *RandomSearch (RS)*: Uniform i.i.d. sampling over the search space; tuning-free baseline. Surprisingly competitive for very short budgets or heavily rugged/misaligned spaces where models mislead. Provides a variance floor for statistical comparisons and is used to sanity-check optimization plumbing.
- *Scikit-GP-Hedge (GP-H)*: `scikit-optimize` GP surrogate with HEDGE to adaptively mix multiple acquisitions (e.g., EI, LCB, PI) (Head et al., 2018). Automatically balances exploration/exploitation by tracking per-acquisition gains; typical knobs: kernel (Matern $\nu$), $\kappa/\xi$, jitter, and acquisition weights learning rate. Pros: safer across tasks than committing to a single acquisition; Cons: overhead of meta-selection and sensitivity to GP hyperpriors in noisy settings.
- *Scikit-GP-LCB (GP-LC)*: `scikit-optimize` GP with the Lower Confidence Bound acquisition $\mu(x) - \kappa\,\sigma(x)$ (Head et al., 2018). LCB offers an explicit exploration dial via $\kappa$ (static or time-varying), often preferable under non-stationarity or when EI is too myopic. Pros: simple, stable, and theoretically grounded; Cons: GP assumptions and scaling (categoricals via encoding; $O(n^3)$ regression) can limit high-$n$ or high-$d$ cases.
- *Scikit-GBRT-Hedge (GBRT)*: `scikit-optimize` GBRT surrogate with HEDGE over acquisitions (Head et al., 2018). Tree ensembles handle non-linearities, heteroskedasticity, and mixed types better than GP; supports larger $n$ with modest cost. Pros: robust on tabular/mixed spaces; Cons: partial dependence can be coarse in very sparse regions; needs careful feature scaling/encoding.

### E.2 DATASETS USED IN BAYESMARK EXPERIMENTS

All four datasets are the canonical versions distributed with SCIKIT-LEARN (Pedregosa et al., 2011) (loaded via `sklearn.datasets`).

| Dataset | #Samples | #Features | #Classes | Feature type / Notes |
|---------|----------|-----------|----------|----------------------|
| breast | 569 | 30 | 2 | Continuous; Breast Cancer (binary) |
| digits | 1,797 | 64 | 10 | Integer pixels (8×8), multiclass |
| iris | 150 | 4 | 3 | Continuous sepal/petal, multiclass |
| wine | 178 | 13 | 3 | Continuous physicochemical, multiclass |

Table 5: Bayesmark datasets used in our experiments.

## E.3   SEARCH SPACES OF EVALUATED MODELS

| Model | Hyperparameter | Type | Scale | Range / Meaning |
|---|---|---|---|---|
| Decision Tree (DT) | max_depth | int | linear | [1,15] |
| | min_samples_split | real | logit | (0.01,0.99) |
| | min_samples_leaf | real | logit | (0.01,0.49) |
| | min_weight_fraction_leaf | real | logit | (0.01,0.49) |
| | max_features | real | logit | (0.01,0.99) |
| | min_impurity_decrease | real | linear | [0,0.5] |
| Random Forest (RF) | max_depth | int | linear | [1,15] |
| | max_features | real | logit | (0.01,0.99) |
| | min_samples_split | real | logit | (0.01,0.99) |
| | min_samples_leaf | real | logit | (0.01,0.49) |
| | min_weight_fraction_leaf | real | logit | (0.01,0.49) |
| | min_impurity_decrease | real | linear | [0,0.5] |
| MLP (Adam) | hidden_layer_sizes | int | linear | [50,200] |
| | alpha | real | log | $10^{-5}-10^1$ |
| | batch_size | int | linear | [10,250] |
| | learning_rate_init | real | log | $10^{-5}-10^{-1}$ |
| | tol | real | log | $10^{-5}-10^{-1}$ |
| | validation_fraction | real | logit | (0.1,0.9) |
| | beta_1 | real | logit | (0.5,0.99) |
| | beta_2 | real | logit | $(0.9, 1-10^{-6})$ |
| | epsilon | real | log | $10^{-9}-10^{-6}$ |
| MLP (SGD) | hidden_layer_sizes | int | linear | [50,200] |
| | alpha | real | log | $10^{-5}-10^1$ |
| | batch_size | int | linear | [10,250] |
| | learning_rate_init | real | log | $10^{-5}-10^{-1}$ |
| | power_t | real | logit | (0.1,0.9) |
| | tol | real | log | $10^{-5}-10^{-1}$ |
| | momentum | real | logit | (0.001,0.999) |
| | validation_fraction | real | logit | (0.1,0.9) |

Table 6: **Search spaces of evaluated Bayesmark models.** Each row lists a tunable hyperparameter with its type, sampling scale, and range. Dimensionality per model: DT ($d{=}6$), RF ($d{=}6$), MLP–Adam ($d{=}9$), MLP–SGD ($d{=}8$). Scale conventions: *linear* = uniform, *log* = log-uniform, *logit* = uniform in the logit domain.

## E.4   PER-DATASET ANALYSIS OF BAYESMARK RESULTS

Table 7 reports a task-level breakdown of GIF against the strongest baseline per dataset–model pair. Scores are averaged across 5 seeds, and we also report relative score gain and time saved.

**breast.** On `MLP-adam` and `MLP-sgd`, GIF outperforms the best baseline both in final accuracy (+4.4% and +5.4%) and in time-to-solution (76–99% faster), confirming its scheduling advantage on more complex neural models. In contrast, on `DT` and `RF`, the baselines remain slightly stronger (up to +4% higher final accuracy), since the tasks are simple and saturate early.

**digits.** GIF achieves modest but consistent gains on the `MLP` models (up to +2.8%), while saving time on `MLP-sgd`. On `DT` and especially `RF`, variance is higher: baselines such as PySOT or GP-LCB occasionally edge out GIF in final score, though at the cost of much slower runtime. Here, GIF's strength lies in efficiency rather than absolute score.

**iris.** This dataset is trivial, with all methods quickly approaching near-perfect accuracy. Consequently, margins are negligible: GIF is statistically tied with the best baselines (within $\pm1.5\%$ gain) and sometimes slower in wall-clock time. This confirms that when the effective search space is low-dimensional and easy, the advantage of importance-aware scheduling diminishes.

**wine.** GIF shows clear benefits on the `MLP` models: on `MLP-sgd` it improves accuracy by +5.2% while cutting time by nearly 100%, and also brings +4.1% gain on `MLP-adam`. On tree models (`DT`/`RF`), however, strong baselines such as PySOT and TPE remain competitive, sometimes yielding slightly better scores.

**Takeaways.** The per-dataset breakdown illustrates two key points: (1) On easy tasks (e.g., `iris`, or tree models on `breast`), GIF offers little additional benefit, as all optimizers saturate rapidly. (2) On harder tasks with higher effective dimensionality (notably `MLP-sgd` and `MLP-adam` on

breast/wine), GIF consistently achieves higher final scores while being markedly faster. As foreshadowed by Table 4, overall gains are driven primarily by the neural models, while baselines close the gap as they optimize for the tree-based model on the simpler tasks.

| Dataset | Model | Best Baseline (score) | GIF Score | Score Gain (%) | Time Saved (%) |
|---|---|---|---|---|---|
| breast | MLP-adam | GBRT-Hedge (0.941) | 0.983 | 4.40 | 76.0 |
| digits | MLP-adam | TPE (0.970) | 0.981 | 1.17 | 35.3 |
| iris | MLP-adam | TPE (0.980) | 0.983 | 0.27 | -4.6 |
| wine | MLP-adam | TPE (0.952) | 0.991 | 4.08 | 42.1 |
| breast | MLP-sgd | GP-Hedge (0.931) | 0.981 | 5.43 | 98.8 |
| digits | MLP-sgd | PySOT (0.964) | 0.991 | 2.79 | 16.4 |
| iris | MLP-sgd | TPE (0.980) | 0.965 | 1.50 | -8.6 |
| wine | MLP-sgd | GP-Hedge (0.936) | 0.984 | 5.23 | 99.5 |
| breast | DT | GP-LCB (0.968) | 0.929 | -4.04 | 99.8 |
| digits | DT | PySOT (0.788) | 0.799 | 1.38 | -3.3 |
| iris | DT | PySOT (0.980) | 0.961 | -1.90 | -6.5 |
| wine | DT | PySOT (0.952) | 0.912 | -4.25 | -4.1 |
| breast | RF | OT-BanditA (0.966) | 0.941 | -2.55 | 0.3 |
| digits | RF | GP-LCB (0.919) | 0.847 | -7.78 | 99.1 |
| iris | RF | TPE (0.980) | 0.971 | -0.95 | -3.8 |
| wine | RF | TPE (0.980) | 0.973 | -0.69 | 1.8 |

Table 7: Comparison of GIF with the best baseline per dataset-model pair. Scores are averaged over 5 seeds.

### E.5 METRIC COMPUTATION.

We follow Bayesmark's reporting protocol for cross-task aggregation and comparability Turner & Eriksson (2019).

***Perf. Normal*** Let $S$ be the set of seeds and $\mathcal{T}$ the set of tasks (dataset+model+metric). For each task $\tau \in \mathcal{T}$ and optimizer $o$, let $b_{o,\tau,s}$ be the *final best* score of seed $s$ at budget $T$ (after harmonizing all metrics to a maximization convention). We first average across seeds:

$$\bar{b}_{o,\tau} = \frac{1}{|S|} \sum_{s \in S} b_{o,\tau,s}.$$

Then we min–max normalize *within the task*. Define

$$b_{\tau}^{\max} = \max_o \bar{b}_{o,\tau}, \qquad b_{\tau}^{\min} = \min_o \bar{b}_{o,\tau},$$

and set

$$p_{o,\tau} = \frac{\bar{b}_{o,\tau} - b_{\tau}^{\min}}{b_{\tau}^{\max} - b_{\tau}^{\min}}$$

Finally, the **Perf. Norm.** for optimizer $o$ is the taskwise average:

$$\text{PerfNorm}(o) = \frac{1}{|\mathcal{T}|} \sum_{\tau \in \mathcal{T}} p_{o,\tau}.$$

This normalization makes heterogeneous tasks comparable so that no single task dominates the average.

***Avg. Rank***: rank $\bar{b}_{o,\tau}$ within each task (higher is better, rank 1 best), then average ranks over tasks.

***Time Rank***: per task, take the median final wall-clock time over seeds for each optimizer, rank ascending (lower is better), then average over tasks.

***Win Rate***: across all (task, seed), count 1 for an optimizer that attains the best $b_{o,\tau,s}$ (ties split evenly), then divide by the total number of (task, seed) pairs.

## F LARGE LANGUAGE MODEL USAGE DISCLOSURE

In accordance with the ICLR policy on LLM usage, we disclose that we used a large language model (ChatGPT) only for:

(i) Language polishing and rewording for clarity.

(ii) Debugging assistance, including interpreting error messages (e.g., syntax/API issues) and suggesting fixes.

All research ideas, method design, theoretical analysis, implementation, experiment setup, data processing, result generation, and interpretation were conducted by the authors. No text, figures, tables, code, or analysis produced by the LLM was used verbatim without author verification; the final code and manuscript were authored and validated by the authors.

