# OpenReview forum: "Greedy Importance First (GIF): Importance-Aware Scheduling for Hyperparameter Optimization"
_ICLR.cc/2026/Conference — ICLR 2026 Conference Withdrawn Submission_

### Official Review · Reviewer_VBHb · 2025-10-26

**Soundness:** 2
**Presentation:** 2
**Contribution:** 2
**Rating:** 2
**Confidence:** 4

**Summary:**

The paper proposes Greedy Importance First as a scheduling strategy for HPO.
The goal is to improve sample efficiency of HPO in high-dimensional anisotropic search spaces.
GIF uses a small-sample warm start on sub-sampled data to perform a hyperparameter importance assessment (e.g. using N-RReliefF as a method by default).
The remaining optimization budget is then allocated in rounds by sorting hyperparameters by importance, grouping them, and allocating trials proportionally to the group importance, and performing subspace optimization by fixing non-optimized parameters to the current incumbent.
A full-space fallback is used for when a round yields no improvement.
The authors compare GIF on weighted analytic functions, Bayesmark and NB301 against competitors such as TPE, BOHB (single-fidelity), and random search and show that GIF can perform competitively.

**Strengths:**

The idea itself of using early, empirically derived hyperparameter importance assessment to guide the optimization is somewhat novel and original and an intuitive approach to counteract the potential uneven influence of hyperparameters in a search space.
While GIF makes use of existing building blocks (HIA) for HPO, I can see some contribution here.

The paper in general is written well and code for reproducibility is provided.

GIF shows good performance on NB301.

**Weaknesses:**

Baselines such as local search (https://arxiv.org/abs/2005.02960), TurBO (https://arxiv.org/abs/1910.01739), SMAC (especially SMAC4MF for NB301, https://arxiv.org/abs/2109.09831) were not included (although some of them are mentioned as related work).

Multi-fidelity is not touched upon in the paper. The single-fidelity restriction on BOHB is not justified. The choice of using sub-sampled data for the warm start for HIA, rather than other standard multi-fidelity proxies such as small number of epochs, should also be discussed.

It is not clear why GIF cannot be extended to multi-fidelity (i.e., via HB) and benchmarking full-fidelity on NB301 is not reasonable and convincing, given the training and validation cost of NAS from scratch.

The one-time HIA based only on the small warm-start budget can be vulnerable to noise from the initial samples, especially if the sub-sampled data provides a misleading signal. This is not discussed in the paper (and neither is the choice to use sub-sampled data here instead of other lower fidelity signals, i.e. based on smaller number of epochs for NB301).

Key parameters like the group size and fallback quota are justified only as practical defaults and lack strong empirical evidence of their robustness across problems.

I would argue that using Bayesmark is no longer timely. The HPO community tends to rely on newer benchmarking suites (https://arxiv.org/abs/2109.06716, https://arxiv.org/abs/2109.06716, ...)

Overall, the paper has limited contributions by proposing to make use of HIA for HPO but 1) falls short in extending it to multi-fidelity which is needed for NAS from scratch 2) suffers from missed comparisons against SOTA methods 3) does not convincingly demonstrate that GIF brings strong benefits over baselines on a substantial number of relevant benchmarks.

**Questions:**

1. Given that HPO for modern deep learning relies on multi-fidelity, please clarify how GIF would combine with a standard HB loop. Would it not make sense to reformulate GIF here as a multi-fidelity method and then compare against actual multi-fidelity methods (BOHB, SMAC4MF, ...) also in benchmarks?

2. Can the authors justify the decision to perform HIA only on the warm start rather than updating it? What is the computational overhead of N-RReliefF relative to a single full-fidelity evaluation on NB301?

3. Can the authors compare against BOHB, SMAC4MF in actual multi-fidelity settings (at least on NB301)? Would it be possible to compare also against local search and TurBO (especially on the weighted analytic functions)?

4. Please provide a justification why HIA is performed only once. Can you discuss the risk of noise/instability and the computational trade-off?

5. What exactly are the benefits of N-RReliefF as an HIA method compared to e.g., fANOVA?

---

### Official Review · Reviewer_5gg1 · 2025-10-29

**Soundness:** 1
**Presentation:** 3
**Contribution:** 2
**Rating:** 2
**Confidence:** 4

**Summary:**

The paper studies how to improve HPO, knowing that only a few dimensions are relevant. The proposed method, GIF, runs an initial design and then assesses the hyperparameters' importance. Based on these results, it splits the search space into smaller sub-search spaces, where the "unimportant" hyperparameters are kept fixed to the incumbent values. It runs regular HPO in these smaller spaces until further improvement is found. By iterating between optimizing only the important hyperparameters and the whole space, GIF can improve on existing HPO methods. The paper evaluates GIF on several benchmark functions and shows the importance of design decisions via an ablation analysis.

### Minor comments that did not influence my rating:
  * The paper introducing Fanova for hyperparameter importance already studied how to use importance results to speed up hyperparameter search. A similar setup was discussed later in the work. Both should be addressed as related work:
   [1] Hutter, F., Hoos, H., Leyton-Brown, K.. (2014). An Efficient Approach for Assessing Hyperparameter Importance. ICML 2014
   [2] Jan N. van Rijn and Frank Hutter. (2018). Hyperparameter Importance Across Datasets. KDD 2018
  * BOHB is a multi-fidelity method. Thus, its bad performance compared to TPE is surprising. It should perform better than TPE as it uses TPE as the underlying method and _also_ leverages multi-fidelity optimization. Furthermore, it is unclear how the multi-fidelity setups work for the synthetic functions.
  * Line 136 and line 152. I would not consider TPE a "traditional" BO method. To understand the explanation of TPE, one would need to know how vanilla BO with a GP and an acquisition function works.

**Strengths:**

[significance] addresses a well-known issue (how to leverage knowledge about the low intrinsic dimensionality of HPO tasks) for improving HPO
[impact] general-purpose method that could be applied beyond HPO tasks

**Weaknesses:**

I have three main concerns about this work:

[1] Validity of importance results. While the initial design might provide an unbiased estimate of importance values (the samples could be i.i.d.), follow-up importance calculations might be biased since the observations stem from HPO runs in the subspace. The paper does not discuss whether the importance method works robustly under these conditions.

[2] Missing baselines and relevance of HPO tasks. Several established Bayesian Optimization methods have been developed explicitly for high-dimensional problems, yet they are not included in the experimental evaluation. Although the paper references some approaches, such as TuRBO [1] and "Vanilla" BO [2], other relevant methods, including REMBO [3] and its extensions, are omitted. Without such baselines, the empirical results are not convincing, and the conclusions are limited. Furthermore, the relevance of the benchmark tasks for current ML workflows is limited: Bayesmark optimizes the hyperparameters of standard ML methods on toy datasets like iris, and NASBench301 (the related paper has been published at ICLR and not arXiv) is a highly categorical search space (a task characteristic that is not at all discussed in the paper).

[3] Unclear design choices. This method has two main ingredients: (a) the hyperparameter importance method and (b) the optimization method. It remains unclear whether alternative methods, such as fANOVA for importance estimation or more advanced Bayesian Optimization variants for optimization, could perform equally well or better. A framework agnostic to the choice of these components could strengthen the contribution, but this point would need to be discussed and evaluated.

## Reference
[1] David Eriksson, Michael Pearce, Jacob R Gardner, Ryan Turner, and Matthias Poloczek. 2019. Scalable global optimization via local Bayesian optimization. NeurIPS 2019
[2] Carl Hvarfner, Erik O. Hellsten, Luigi Nardi. Vanilla Bayesian Optimization Performs Great in High Dimensions. ICLR 2024
[3] Ziyu Wang, Frank Hutter, Masrour Zoghi, David Matheson, Nando de Freitas. Bayesian Optimization in a Billion Dimensions via Random Embeddings. JAIR 2016

**Questions:**

See my concerns under the weakness section on [1] biased samples and [3] design choices.

---

> ### Comment · Reviewer_5gg1 · 2025-11-25
>
> (for completeness) since there is no response by the authors, I will not change or update my review/score

---

### Official Review · Reviewer_DsPA · 2025-10-29

**Soundness:** 2
**Presentation:** 3
**Contribution:** 2
**Rating:** 2
**Confidence:** 5

**Summary:**

The authors of the paper propose to use hyperparameter importance assessment to shrink the search space for HPO (and NAS) and assign optimization budget depending on the importance of a hyperparameter. To this end, they apply N-RReliefF and combine it with TPE. In the experiments, they study the performance on artificial functions, Bayesmark and NASBench-301.

**Strengths:**

It is a common (and already fairly old) observation that the effective dimensionality of many hyperparameter optimization problems is fairly small, but the most important hyperparameters depend on the dataset at hand. There not many methods so far exploiting this fact and improving HPO performance by that. GIF is a novel method in this direction.

The performance of GIF on the chosen benchmarks is overall strong and shows the potential of the approach. In combination with the shown ablation study, several design decisions of GIF seem to be reasonable.

**Weaknesses:**

First of all, I wonder a bit about the motivation of the paper. If a subset of the hyperparameters is unimportant, why does it matter for the optimization? By definition of being unimportant, it does not interact with other hyperparameters and can be set to arbitrary values. I’m missing a better argumentation of why this approach is reasonable in the first place.


Secondly, the paper discusses the idea in very broad terms, but then focuses quite a bit on TPE as the underlying optimizer and N-RRelief as the hyperparameter importance approach. I wonder whether the same approach can also be applied to standard BO (according to the claim of the authors of proposing a general approach) and how dependent it is on N-RRelief (which is not an established method).


I also don’t understand the motivation behind assigning optimization budget to subsets of hyperparameters that are deemed unimportant. This should be a waste of computing time by definition. (Section 4.4)


I’m missing at least a reference ot “Random Search for Hyper-Parameter Optimization” by Bergstra and Bengio 2012 as one of the first papers showing that there is low-effective dimensionality in HPO.


The experiments show a fairly consistent picture, but also have several shortcomings:
* 5 seeds are in general very little in view of the highly non-deterministic nature of HPO methods. I would recommend at the very least 10. 50 would be best.
* At the beginning of Section 5, several parameters of GIF are documented (which I appreciate in general). However, any justification for why these were set this way is missing. The statement that “these values were chosen as reasonable defaults” is very hand-wavy.
* Bayesmark was already, at the moment of release, a fairly trivial (and outdated) HPO benchmark. YAHPOGym or CARPS provide more interesting benchmarks.
* It is surprising to me that GIF identifies important hyperparameters very well for d=5, but fails to perform well on these benchmarks (see appendix). The authors are not openly discussing this in Section 6.2.
* In Section 6.4, the authors show results on NASBench-301, which indicates that GIF does not perform better than other methods in the beginning but can only achieve better performance at the very end of the trajectory. The authors emphasize that the full-space fall-back helps (Lines 449-450) in the late phase. This does not make much sense to me because the subspace (which is the main selling point of the paper) does not work well in the beginning but the full space at the end (which all other methods also use throughout the whole optimization) is doing the trick? Counterintuitive to me and not well discussed by the authors.
* I’m missing stronger baselines for high-dimensional BO (at the very least TURBO). Optimally, I would be interested in a combination of TURBO and GIF since the authors claim that their approach is orthogonal to other high-dimensional BO approaches.
* Last but not least, I would be interested to know whether these benchmarks indeed exhibit the characteristics of a low-effective dimensionality.

**Questions:**

1. Why should we spend optimization budget on unimportant hyperparameters?
2. What exactly is GIF measuring? It sounds like it could also assign high importance to hyperparameters exhibiting a very rough, bumpy optimization landscape? I’m not sure whether this is wanted.
3. How are interaction effects considered? It can happen that the main effect of hyperparameters is fairly small but the interaction effects has a strong impact on performance.

---

> ### Comment · Reviewer_DsPA · 2025-11-25
>
> Similar to 5gg1 and with the strong consistency in the ratings of other reviewers, I do not see a reason to change my review.

---

### Official Review · Reviewer_z4eV · 2025-10-29

**Soundness:** 3
**Presentation:** 2
**Contribution:** 2
**Rating:** 2
**Confidence:** 4

**Summary:**

The paper addresses the problem of high-dimensional hyperparameter optimization (HPO) problems where different hyperparameter dimensions affect performance improvement differently, more specifically, anisotropic high-dimensional search spaces.
Greedy Importance First (GIF) is introduced as a solution that performs budget allocation over the function evaluations, scheduling dynamic groups of hyperparameter subspaces based on their estimated relative importance so far.
This can, in principle, be plugged into any Bayesian Optimization (BO) loop to bias the data-driven exploration of the space.
The empirical evidence suggests such greedy scheduling improves performance over its vanilla counterparts.

**Strengths:**

* Clear scope and story, easy to read and follow for the most part, some details aside.
* Carefully designed algorithm with multiple components and thus notations, including details in the Appendix that flow well.
* Adequate coverage of empirical evaluations with synthetic to tabular and surrogate benchmarks, and some well-established baselines for hyperparameter optimization.

**Weaknesses:**

* Many minute details are hard to pick up when considering the full GIF pipeline. Not all such design choices are adequately motivated or justified with ablation experiments. Not all design choices and pseudo-code are necessarily clear. (Please see, *Questions* below)
* Given the use of TPE as the algorithm for the contributed GIF algorithm, it is unclear what the differences are with the only sophisticated, similar baseline in Sequential Grouping (SG), and why it might be worse than Random Search and TPE for 200 function evaluations (Fig. 3).
  * More generally, SG should be explained more in the main or the appendix.
* No strong justification for not including any subspace selection-based high-dimensional BO methods (L87-88): given the GIF algorithm's budget scheduling effectively enforces subspace optimization and is thus not entirely *orthogonal* (L92).

**Questions:**

Below is a combined enumeration of questions and suggestions.

Please note that the rating would go up, especially if these are clarified:
* Difference with SG (see point 6)
* Choice of baselines: high-dim BO baselines (see points 1, 11)
* Clarification on the choice and use of BOHB in the experiments (see points 2, 12)

1\. L92: the *orthogonality* of the approach seems debatable; if the *how* of grouping or subspacing is ignored, some algorithms can still allow for this method to be plugged in with different surrogates and acquisitions for BO. What would the authors say to clarify this here or in the manuscript? This is aside from the novelty aspect coming from the budget scheduling.

2\. L105-110: It is unclear how exactly different fidelity choices are compatible with the function evaluation level budget allocation in GIF. Especially when different gray-box algorithms have their internal discretization crucial to their correctness.

2\.1 BOHB's original implementation (and algorithm) began sampling from the surrogate (acquisition) only after $d+1$ samples were collected, at each fidelity level. Until then it does Random Search (RS). It is totally unclear how the $b_j$ allocated to $G_j$ for a BOHB run adequately allows efficient model-based sampling to kick-in. Note, this suggest BOHB's own hyperparameters are important and should be detailed with the benchmark).

2\.2. Could the above explain why BOHB and RS seem to have the same runtime in Figure 5 (right)? Fig. 10 (bottom), BOHB is too uncannily similar to RS.

3\. L12-118: [1] could also be cited here already.

4\. All subsections in Section 4 could cite suitable Section 5.1.1.

5\.1. Section 4.2: Would not a simpler random search on subspaces (or random importance) yield a better calibrated importance scoring? What do the authors think about this?

5\.2. What is the motivation (and empirical effect) of the subsampling of datasets in the Warm-Start stage?

6\. Sequential Grouping (SG):

6\.1. Given the recency of the method, it would be beneficial for the readers to have some description of this method.

6\.2. On skimming the referred paper (L293), it reads quite similar to the proposed method GIF and thus further warrants more explanation of SG, especially where GIF differs.

6\.3. Assuming 6.2 is true, SG performing worse than RS for nearly 200 evaluations (Fig. 3) either needs more explanation or adequate justification for SG's performance as the closest baseline in the list.

6\.4. Figure 5 (right): Could the authors explain what SG does and how it is faster than even Random Search?

7\. Table 3 and 4 could be made consistent in terms of legibility.

8\. Figure 4: Hard to read plot; could make a legend instead of a title string; declutter the points with different markers and its own separate legend.

9\. Section 7: Could include a *Limitations* section, which could feature or refer to contents in the Appendix (For space, L275-291 could be in the Appendix).

10\. L294: What is the acquisition function used under `BO with GPs`?

11\.1. Could the authors clarify which baselines share the same warmstart history? Should there be two separate evaluations: GIF vs baselines-with-warmstart and GIF vs baselines-without-warmstart?

11\.2. Could the authors explain why even one specialized high-dim BO baseline is not necessary for the GIF argument?

12\.1. Could the authors provide more clarification on the hyperparameters/inputs to GIF (Alg. 1)?

12\.2. How should they vary based on the dimensionality of the problem or possible choice of $A_\text{opt}$?

12\.3. The budget efficiency of GIF in practice appears to be dependent on $B_\text{step}$, which, together with $d$ would define the group sizes and thereby the exact budget allocation. What would the authors recommend for this value?

12\.4. How does 12\.3 influence the gray-box algorithm fidelity choices and also the choice of $A_\text{opt}$ as a function of $B_\text{step}$ (or vice versa)?


---

References:

[1] $\pi$BO: Augmenting Acquisition Functions with User Beliefs for Bayesian Optimization, Hvarfner et al. 2022.

---

### Note · Authors · 2026-01-20

I have read and agree with the venue's withdrawal policy on behalf of myself and my co-authors.